# LRRTM1 underlies synaptic convergence in visual thalamus

Aboozar Monavarfeshani[1,2], Gail Stanton[1,3], Jonathan Van Name[1], Kaiwen Su[1], William A Mills III[1,4], Kenya Swilling[1], Alicia Kerr[1,4], Natalie A Huebschman[5], Jianmin Su[1], Michael A Fox[1,2,3]*

[1]Developmental and Translational Neurobiology Center, Virginia Tech Carilion Research Institute, Roanoke, United States; [2]Department of Biological Sciences, Virginia Tech, Blacksburg, United States; [3]Virginia Tech Carilion School of Medicine, Roanoke, United States; [4]Translational Biology, Medicine, and Health Graduate Program, Virginia Tech, Blacksburg, United States; [5]Roanoke Valley Governor School, Roanoke, United States

**Abstract** It has long been thought that the mammalian visual system is organized into parallel pathways, with incoming visual signals being parsed in the retina based on feature (e.g. color, contrast and motion) and then transmitted to the brain in unmixed, feature-specific channels. To faithfully convey feature-specific information from retina to cortex, thalamic relay cells must receive inputs from only a small number of functionally similar retinal ganglion cells. However, recent studies challenged this by revealing substantial levels of retinal convergence onto relay cells. Here, we sought to identify mechanisms responsible for the assembly of such convergence. Using an unbiased transcriptomics approach and targeted mutant mice, we discovered a critical role for the synaptic adhesion molecule Leucine Rich Repeat Transmembrane Neuronal 1 (LRRTM1) in the emergence of retinothalamic convergence. Importantly, LRRTM1 mutant mice display impairment in visual behaviors, suggesting a functional role of retinothalamic convergence in vision.
DOI: https://doi.org/10.7554/eLife.33498.001

*For correspondence:
mafox1@vtc.vt.edu

**Competing interests:** The authors declare that no competing interests exist.

## Introduction

Over thirty classes of functionally and morphologically distinct retinal ganglion cells (RGCs) exist in mammals, each responsible for conveying different features of the visual world and each with unique projections to retinorecipient nuclei within the brain (*Sanes and Masland, 2015*; *Martersteck et al., 2017*; *Baden et al., 2016*). As a group, RGCs innervate over 40 retinorecipient brain regions (*Morin and Studholme, 2014*; *Monavarfeshani et al., 2017*). However, only a subset of RGCs (~50%) innervate relay cells in the visual thalamus (i.e. the dorsal lateral geniculate nucleus [dLGN]) and provide the principal pathway for image-forming visual information to reach the cerebral cortex (*Dhande et al., 2015*; *Seabrook et al., 2017*) (*Figure 1A*). The recent development of transgenic tools to label these classes of RGCs has revealed that their inputs are segregated into distinct class-specific sublamina within visual thalamus (*Huberman et al., 2008*; *Monavarfeshani et al., 2017*; *Huberman et al., 2009*; *Kay et al., 2011*; *Kim et al., 2008*, *2010*; *Hong and Chen, 2011*), supporting the longstanding belief that different features of the visual field are transmitted through the subcortical visual system in parallel, unmixed anatomical channels (*Dhande et al., 2015*; *Cruz-Martín et al., 2014*).

In addition to being segregated based on class, retinal projections in dLGN are unique in that they form structurally and functionally distinct synapses compared to their counterparts in other retinorecipient nuclei (*Hammer et al., 2014*). Retinal terminals in dLGN are prototypic 'driver' inputs which are large (compared to adjacent non-retinal inputs) and capable of generating strong

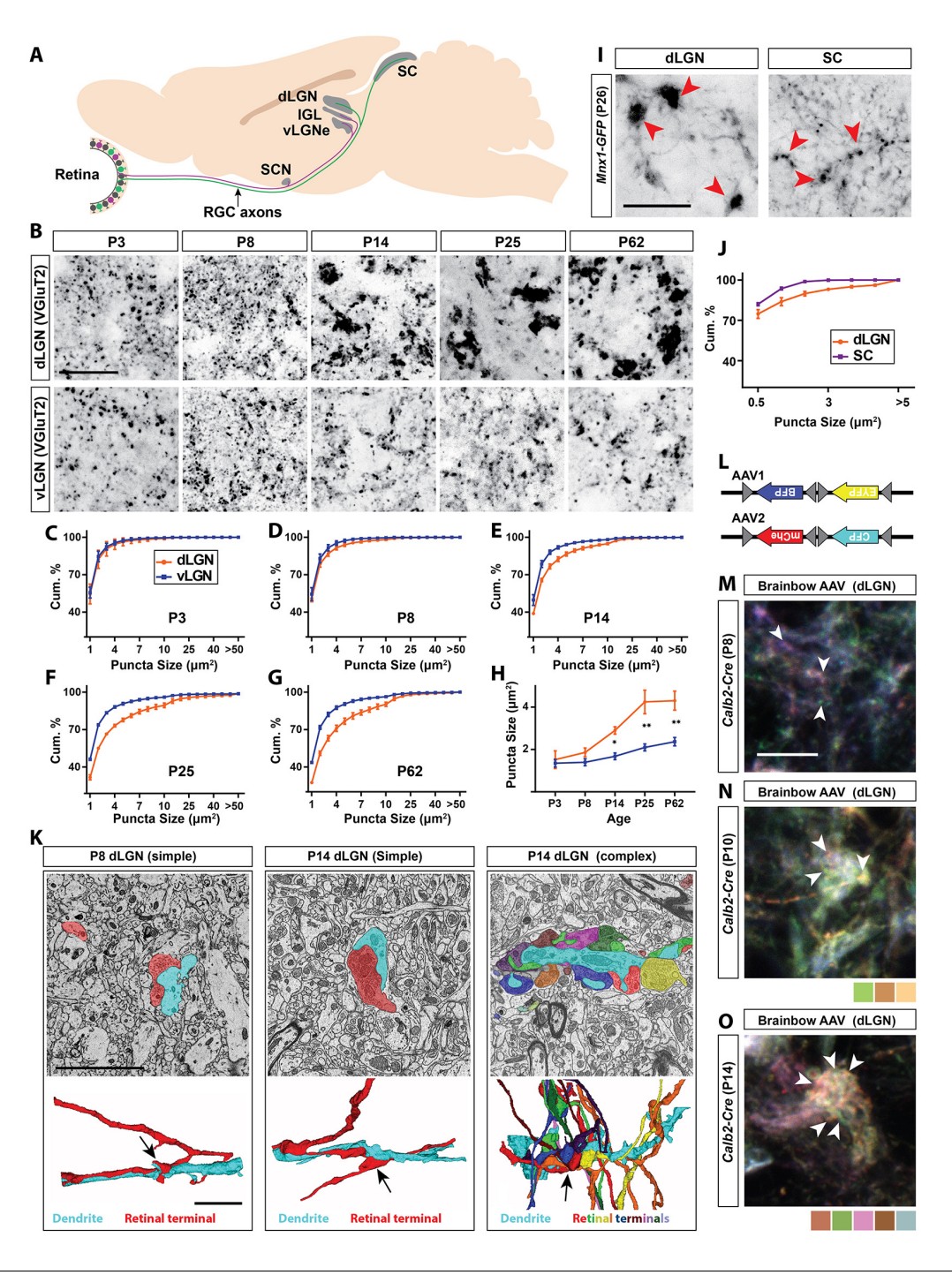

**Figure 1.** Retinal projections develop into unique terminal types in dLGN. (**A**) Schematic of the mouse brain highlighting the main retinorecipient regions including dLGN. (**B**) Development of VGluT2$^+$ retinal terminals in dLGN and vLGN in wild type mice. (**C–G**) Cumulative (cum.) distribution of VGluT2$^+$ puncta size in P3 (**C**), P8 (**D**), P14 (**E**), P25 (**F**) and P62 (**G**) dLGN (orange) and vLGN (blue). Data are shown as Mean ±SEM. (**H**) Average total VGluT2$^+$ terminal size in developing dLGN and vLGN. Data are shown as Mean ±SEM, *p<0.0001 by ANOVA. (**I**) GFP$^+$ retinal terminals in dLGN and superior colliculus (SC) of *Mnx1-GFP* mice. Red arrowheads highlight gfp-labeled retinal terminals. (**J**) Cumulative (cum.) distribution of GFP$^+$ puncta size in P25 dLGN (orange) and SC (purple). Data are shown as Mean ±SEM. (**K**) SBFSEM of retinogeniculate synapses in dLGN of P8 and P14 mice. 3D reconstructions of retinal terminals and relay cell dendrites are depicted below each micrograph. The black arrows denote simple retinogeniculate synapses in P8 and P14 or clusters of retinal terminals originating from

*Figure 1 continued on next page*

*Figure 1 continued*

multiple RGCs in P14 mouse dLGN. (**L**) Schematic representation of brainbow-AAV constructs. (**M–O**) Examples of brainbow-labeled clusters of retinal terminals in dLGN of P8 (**M**), P10 (**N**) and P14 (**O**) *Calb2-Cre* mice. Arrowheads denote terminals labeled by different colors. Scale bars, 20 µm (**B and I**), 5 µm (**K**), 10 µm (**M**).

DOI: https://doi.org/10.7554/eLife.33498.002

The following figure supplement is available for figure 1:

**Figure supplement 1.** Unique transformation of retinal nerve terminals in the developing dLGN.
DOI: https://doi.org/10.7554/eLife.33498.003

excitatory postsynaptic responses in thalamic relay cells. Until recently, it was thought that the level of convergence of retinal inputs onto these relay cells was exceptionally low with only a few (1-5) RGCs innervating each relay cell (*Chen and Regehr, 2000*; *Jaubert-Miazza et al., 2005*; *Sincich et al., 2007*; *Hamos et al., 1987*; *Cleland and Lee, 1985*; *Cleland et al., 1971*; *Mastronarde, 1992*; *Usrey et al., 1999*; *Yeh et al., 2009*; *Weyand, 2016*; *Rathbun et al., 2016*, *2010*). This low level of retinal convergence allows relay cells to faithfully transfer information from RGCs to visual cortex in an unaltered form, also adding support to the notion that information regarding different features of the visual field flow through the thalamus in parallel channels.

Recently, however, a series of anatomical studies in mice have challenged the concept of feature-specific, parallel visual channels by revealing a level of retinal convergence onto relay cells that is more than an order of magnitude higher than previously described (*Hammer et al., 2015*; *Morgan et al., 2016*; *Rompani et al., 2017*; *Howarth et al., 2014*). Not only is there a high level of retinogeniculate (RG) convergence in mice, but some relay cells receive input from functionally distinct classes of RGCs (*Rompani et al., 2017*) raising new questions about the role of thalamus in processing visual information before it reaches visual cortex.

Part of this newly appreciated retinal convergence stems from a set of unique RG synapses (termed complex RG synapses) that contain numerous retinal axons whose terminals aggregate on shared regions of relay cell dendrites (*Morgan et al., 2016*; *Hammer et al., 2015*; *Lund and Cunningham, 1972*). Complex RG synapses have been reported in both rodents and higher mammals (*Lund and Cunningham, 1972*; *Jones and Powell, 1969*; *So et al., 1985*; *Campbell and Frost, 1987*; *Guillery and Scott, 1971*; *Wilson et al., 1984*). Similar to the more classical simple RG synapses (which contain a single retinal terminal on a given portion of a relay cell dendrite), these complex RG synapses are absent from other retinorecipient regions of brain (*Hammer et al., 2014*) (*Figure 1—figure supplement 1*). Since branches of dLGN-projecting RGCs also innervate other retinorecipient nuclei (*Dhande et al., 2015*), we interpret this to suggest that target-derived signals must be generated in dLGN to pattern the unique transformation of retinal axons into simple and complex RG synapses.

In the present study, we sought to identify such target-derived signals. Using next generation sequencing, we discovered that relay cells in dLGN (but not principal neurons in other retinorecipient nuclei) express Leucine Rich Repeat Transmembrane Neuronal 1 (LRRTM1), a known inducer of excitatory synaptogenesis (*Linhoff et al., 2009*; *de Wit et al., 2009*). Genetic deletion of LRRTM1 led to a loss of complex RG synapses and thus reduced retinal convergence in visual thalamus. While mutants lacking LRRTM1 and complex RG synapse exhibit normal visual acuity and contrast sensitivity, they display impaired performance in a set of more complex visual tasks that require processing multiple distinct elements of the visual field. Taken together, these results not only identify a novel mechanism underlying the establishment of retinal convergence in visual thalamus, but also importantly provide the first insight into the functional significance of complex RG synapses (and, possibly, retinal convergence) in vision.

## Results

### Unique transformation of retinal terminals in dLGN coincides with eye-opening

To examine the emergence of the unique morphology of retinal terminals in developing mouse dLGN, two approaches were applied: retinal terminals were either immunolabeled with antibodies

against vesicular glutamate transporter 2 (VGluT2, a synaptic vesicular component only present in retinal terminals in visual thalamus) (*Hammer et al., 2014*; *Land et al., 2004*), or were anterogradely labeled by intraocular injection of fluorescent-conjugated Cholera Toxin B (CTB) (*Muscat et al., 2003*) (*Figure 1B–H*; *Figure 1—figure supplement 1*). Shortly after their initial formation (P3-P8), VGluT2- or CTB-labeled terminals appeared similar in size and morphology in dLGN and the adjacent retinorecipient ventral lateral geniculate nucleus (vLGN). However, by eye-opening (P12-P14), terminals in dLGN underwent significant enlargement compared to those in vLGN and other (*Hammer et al., 2014*) retinorecipient nuclei (*Figure 1B–H*; *Figure 1—figure supplement 1*). The unique developmental transformation of retinal terminals in dLGN at eye-opening (rather than at their initial formation), suggested that this was not the result of purely cell intrinsic mechanisms in dLGN-projecting classes of RGCs.

To test this hypothesis, we assessed retinal terminals generated by a single class of ON-OFF direction-selective RGCs whose axons branch to innervate both dLGN and superior colliculus (SC) (*Dhande et al., 2015*; *Kim et al., 2010*). This class of RGC is specifically labeled in *Mnx1-GFP* mice (also called *Hb9-GFP*)(*Trenholm et al., 2011*). Despite originating from branches of individual retinal axons, those terminals present in dLGN were dramatically larger than those in SC (0.83 ± 0.1 μm$^2$ in dLGN vs 0.36 ± 0.01 μm in SC, p<0.01 by t-test, n = 3, *Figure 1I,J*). These data suggest that target-derived cues are generated in dLGN around the time of eye-opening to pattern the transformation of retinal terminals.

The approaches described above do not provide the resolution required to differentiate simple and complex RG synapses, therefore, we used serial block-face scanning electron microscopy (SBFSEM) to identify whether both simple and complex RG synapses emerged at eye-opening. While SBFSEM ultrastructural analysis revealed the presence of both simple and complex RG synapses shortly after eye-opening (P14) we were only able to identify simple-like RG synapses prior to eye-opening (at P8) (*Figure 1K*). Moreover, we delivered brainbow AAVs (*Figure 1L*) intraocularly in newborn *Calb2-Cre* mice (in which a large proportion of RGCs express Cre recombinase) to generate multi-colored RGCs and assess the development of complex RG synapses (*Hammer et al., 2015*). Similar to SBFSEM analysis, brainbow AAV-labeling failed to detect clusters of retinal terminals at P8 (*Figure 1M*), but clearly revealed clusters of retinal terminals originating from distinct RGCs as early as P10 and P14 (*Figure 1N,O*). Thus, around eye-opening, dLGN-specific molecular mechanisms must emerge to induce the unique transformation of both simple and complex RG synapses.

## Identification of target-derived synaptic organizing molecules in dLGN

To identify target-derived synaptic organizers present at eye-opening in dLGN (but not other retinorecipient regions), we performed next-generation transcriptome analysis of developing mouse visual thalamus (*Figure 2A*; *Figure 2—source data 1*). We assessed four different developmental time points, two before eye-opening (P3 and P8), and two at (P12) or after (P25) eye-opening (*Figure 1*). Comparing gene expression profiles in both dLGN and vLGN revealed hundreds of differentially and developmentally expressed mRNAs (*Figure 2B*). We focused our attention on a small subset of genes that were significantly enriched in dLGN (compared to vLGN) and whose highest expression coincided with eye-opening and the emergence of simple and complex RG synapses. Two genes with well-established roles in inducing excitatory synapses fit those criteria: *Lrrtm1* and *neuritin 1* (*Nrn1*) (*Figure 2C–E*) (*Linhoff et al., 2009*; *Javaherian and Cline, 2005*). We confirmed the enrichment of these genes at eye-opening in dLGN (but not vLGN) by qPCR, in situ hybridization and western blot (*Figures 2F–I* and *3A–D*; *Figure 3—figure supplement 1A–C*). While the increase in expression of these genes coincides with eye-opening and the onset of experience-dependent transformation of retinal nerve terminals, we did not find a decrease in their expression in the absence of visual inputs (*Figure 3—figure supplement 1F*). In addition to their low expression level in vLGN, it is important to point out that *Lrrtm1* and *Nrn1* mRNAs were either absent or only weakly expressed in other retinorecipient nuclei, such as the SC and suprachiasmatic nucleus (SCN) (*Figure 3E*; *Figure 3—figure supplement 1D,E*). There were, however, significant differences in the distribution of *Lrrtm1* and *Nrn1* mRNAs in other regions of the visual system. *Lrrtm1* was not generated by RGCs (although it was expressed in the INL) or by many cells in primary visual cortex (vCTX) (*Figure 3E,F*), whereas *Nrn1* was robustly expressed by both RGCs and by cells in vCTX (*Figure 3E, F*) (see also *Fujino et al., 2008*; *Nedivi et al., 1998*).

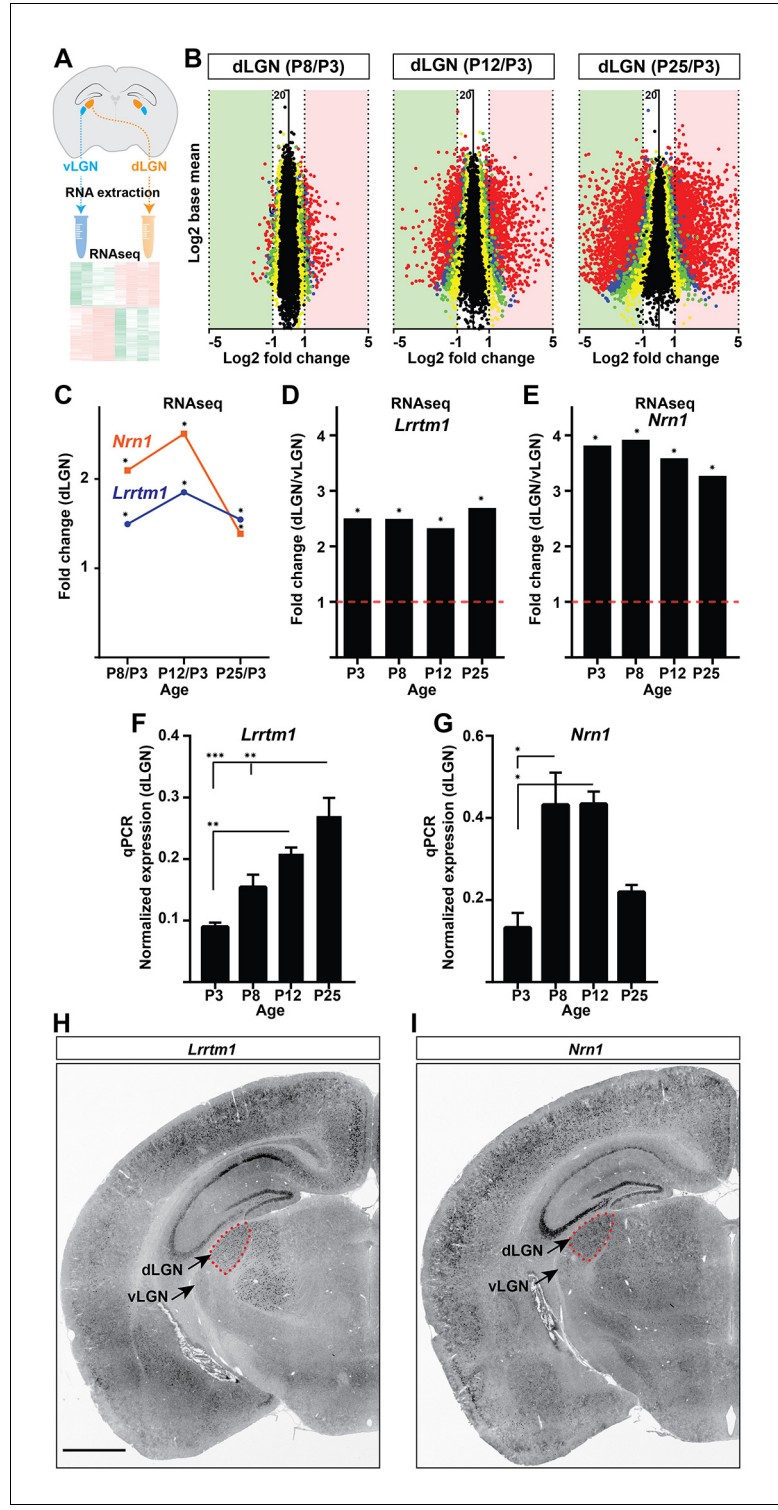

**Figure 2.** Identification of *Lrrtm1* and *Nrn1* as candidate synaptic organizing cues in dLGN. (**A**) Next generation RNAseq was performed on RNA isolated from dLGN and vLGN at P3, P8, P12 and P25. (**B**) Volcano scatter plots show differentially expressed mRNAs in the developing dLGN. (**C**) Relative *Lrrtm1* and *Nrn1* mRNA levels in dLGN at P8, P12 and P25 compared to P3 by RNAseq. Data are relative values comparing different ages, *p<0.0001 by Wald Chi-Squared Test (DESeq2). (**D and E**) Enrichment of *Lrrtm1* (**D**) and *Nrn1* (**E**) mRNAs in dLGN compared to vLGN at four ages in wild type mice. Data are relative values comparing dLGN and vLGN, *p<0.0001 by Wald Chi-Squared Test (DESeq2). (**F and G**) Developmental expression of *Lrrtm1* (**F**) and *Nrn1* (**G**) mRNAs in wild type dLGN
*Figure 2 continued on next page*

*Figure 2 continued*

by qPCR. Data are shown as Mean ±SEM; ***p<0.0001, **p<0.01, *p<0.05 by ANOVA. (H and I) ISH for *Lrrtm1* (F) and *Nrn1* (G) mRNAs in coronal sections of wild type P25 mouse brains. Scale bar, 1 mm (H and I).
DOI: https://doi.org/10.7554/eLife.33498.004
The following source data and figure supplement are available for figure 2:

**Source data 1.**
DOI: https://doi.org/10.7554/eLife.33498.006
**Figure supplement 1.** Expression of NRXN proteins and *lrrtm*, and *nlgn* mRNAs in the subcortical visual thalamus.
DOI: https://doi.org/10.7554/eLife.33498.005

Finally, we sought to address the cell-specific expression of *Lrrtm1* and *Nrn1* in visual thalamus. Since *Lrrtm1* encodes a transmembrane cell adhesion molecule and *Nrn1* encodes a GPI-linked membrane associated extracellular molecule (*Naeve et al., 1997*; *Linhoff et al., 2009*), we hypothesized that dLGN relay cells must generate these molecules for them to act post-synaptically at RG synapses. We combined in situ hybridization analysis using riboprobes against these two genes with molecular and genetic approaches to label different cell types in dLGN. First, we demonstrated that both *Lrrtm1* and *Nrn1* mRNAs are generated by neurons and not glia since they were co-expressed by *Syt1*-expressing neurons (93% of $Syt1^+$ neurons co-expressed *Lrrtm1*; 98% of $Syt1^+$ neurons co-expressed *Nrn1*; *Figure 4A,B*; *Figure 3—figure supplement 1G,H*) but not by IBA1-expressing microglia or GFP-labeled astrocytes in *Aldh1l1-GFP* mice (*Figure 4C–F*). Next, to differentiate which types of neurons generate these synaptogenic cues, we assessed *Lrrtm1* and *Nrn1* mRNA expression in glutamate decarboxylase (GAD67)-expressing inhibitory interneurons and in C*rh-Cre::tdt* transgenic mice in which excitatory thalamocortical relay cells are fluorescently labeled (*Taniguchi et al., 2011*). Results revealed *Lrrtm1* and *Nrn1* are exclusively produced by dLGN relay cells (100% of C*rh-Cre::tdt⁺* cells co-expressed *Lrrtm1* and *Nrn1*; 0% and 2% of $Gad1^+$ cells co-expressed *Lrrtm1* and *Nrn1*, respectively; *Figure 4G–J*; *Figure 3—figure supplement 1I,J*). Based on their developmental and cell-specific expression, these molecules therefore represented prime candidates to influence the development of simple and complex RG synapses.

## LRRTM1 is required for the development of complex RG synapses

Previous studies have reported roles for both LRRTM1 and NRN1 (also called Candidate Plasticity Gene 15, CPG15) in inducing the formation and maturation of excitatory synapses (*Cantallops et al., 2000*; *Ko et al., 2011*; *Linhoff et al., 2009*; *Nedivi et al., 1998*; *Soler-Llavina et al., 2011*). In addition, NRN1 contributes to the development and maturation of retinal arbors (*Cantallops et al., 2000*). To test whether these molecules are necessary for the development of retinal terminals, we assessed the morphology of retinal terminals in dLGN of mice lacking LRRTM1 (*Linhoff et al., 2009*) or NRN1 (*Fujino et al., 2011*) using VGluT2 immunostaining and CTB anterograde labeling. These studies revealed a significant decrease in the number of large VGluT2⁺ and CTB⁺ puncta in dLGN of *Lrrtm1⁻/⁻* mice at and after eye-opening (*Figure 5A–C*; *Figure 5—figure supplement 1A,B,E*), suggesting a role for this molecule in the maturation and/or refinement of RG circuitry. Retinal terminals in neonatal dLGN (i.e. before eye-opening) or in vLGN were not affected by the loss of LRRTM1 (*Figure 5A–F*; *Figure 5—figure supplement 1A–F*). Since retinal projections account for only a small proportion (5–10%) of all projections innervating relay cells residing in dLGN (*Monavarfeshani et al., 2017*), we also assessed whether the loss of LRRTM1 altered other types of terminals in dLGN. None of the non-retinal inputs examined appeared affected in *Lrrtm1⁻/⁻* mutant mice (*Figure 5—figure supplement 1G–I*). Similar analysis in *Nrn1⁻/⁻* mutants failed to identify developmental deficits in the density, size or distribution of retinal terminal in dLGN (*Figure 5—figure supplement 2A,B*).

As described earlier, an important limitation of these techniques is that they cannot differentiate simple or complex RG synapses. It was therefore unclear whether individual retinal terminals were smaller in *Lrrtm1⁻/⁻* mice, or clusters of retinal terminals were absent in these mutants. To overcome this technical limitation, we employed both SBFSEM and multicolor brainbow-AAV labeling of retinal axons in dLGN of *Lrrtm1⁻/⁻* mutant and control mice. In SBFSEM, retinal terminals were distinguished from all other synaptic terminals by their round vesicles and pale mitochondria (*Rafols and*

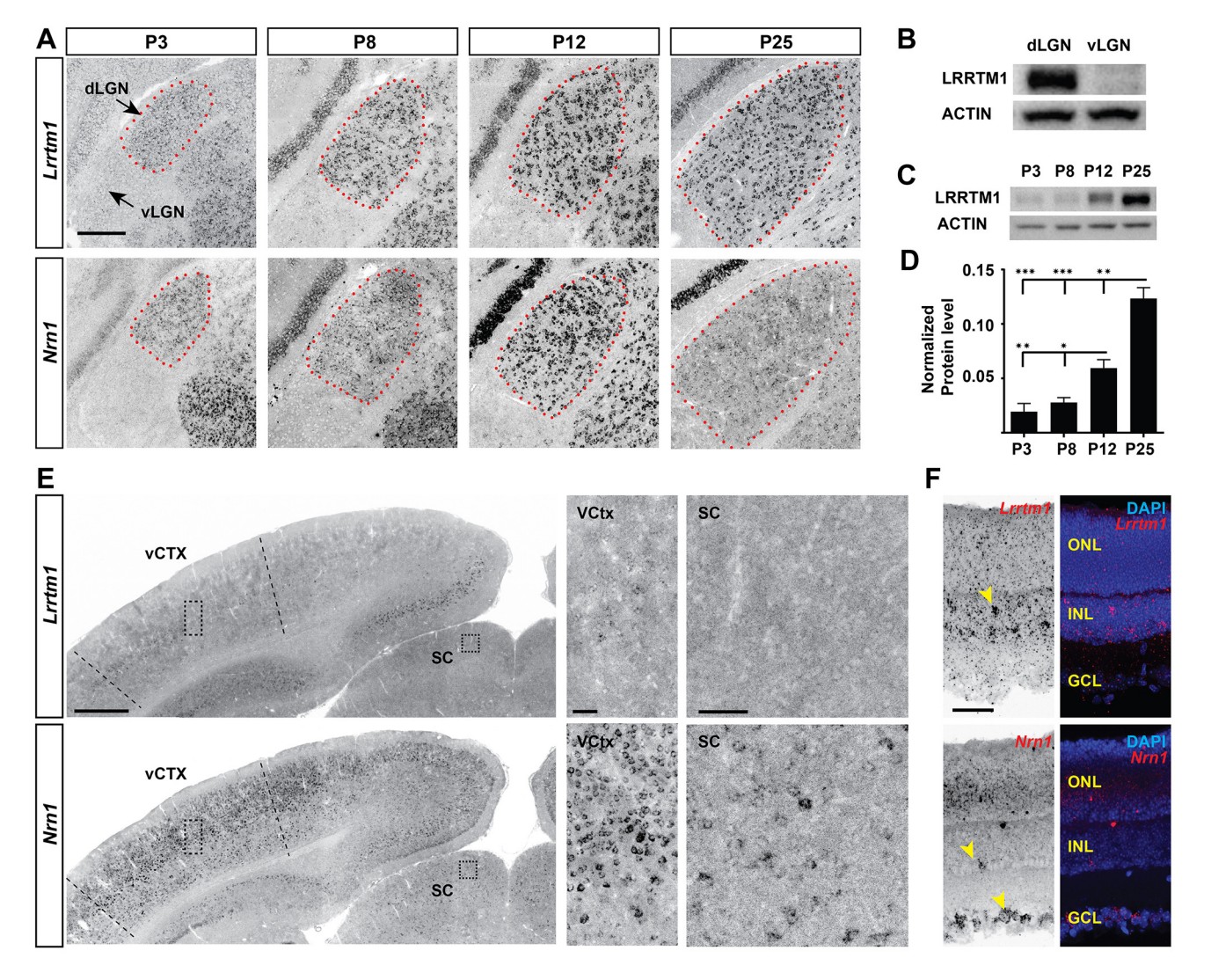

**Figure 3.** Developmental and region-specific expression of *Lrrtm1* and *Nrn1*. (**A**) ISH for *Lrrtm1* and *Nrn1* mRNAs in the developing visual thalamus. dLGN encircled by red dots. (**B–D**) Western blots show LRRTM1 protein level is higher in dLGN than vLGN (**B**) and increases in the dLGN postnatally (**C** and **D**). Data are shown as Mean ±SEM; ***p<0.0001, **p<0.01, *p<0.05 by ANOVA. (**E**) Expression of *Lrrtm1* and *Nrn1* mRNAs in coronal sections of P25 mouse brains. Boxes in visual cortex (vCTX) and superior colliculus (SC) are shown in higher magnifications on the right. (**F**) Expression of *Lrrtm1* and *Nrn1* mRNAs in P25 retina. Yellow arrowheads denote mRNA expression. ONL, outer nuclear layer; INL, inter nuclear layer; GCL, ganglion cell layer. Scale bars, 200 μm (**A**), 500 μm (**E**), 50 μm (insets of vCTX and SC and F).

DOI: https://doi.org/10.7554/eLife.33498.007

The following figure supplement is available for figure 3:

**Figure supplement 1.** Region- and cell-specific expression of *Lrrtm1* and *Nrn1*.
DOI: https://doi.org/10.7554/eLife.33498.008

*Valverde, 1973*; *Hammer et al., 2014*; *Bickford et al., 2010*) and were traced throughout the entire volume of the imaged tissues. In total, 534 retinal terminals were analyzed in *Lrrtm1*$^{-/-}$ mice and 646 in controls (n = 3 mice per genotypes). While the majority (90%) of RG synapses were classified as complex in controls, only 37% of retinal terminals fell into this category in *Lrrtm1*$^{-/-}$ mutant dLGN (*Figure 6A–C*; *Figure 6—figure supplement 1A,B*). Similarly, analysis of multicolor-labeled retinal terminals by brainbow AAVs showed fewer and smaller clustered retinal terminals in dLGN of *Lrrtm1*$^{-/-}$ mice (*Figure 6G–I*). The majority of retinal terminal clusters in mutants (82.6%) contained

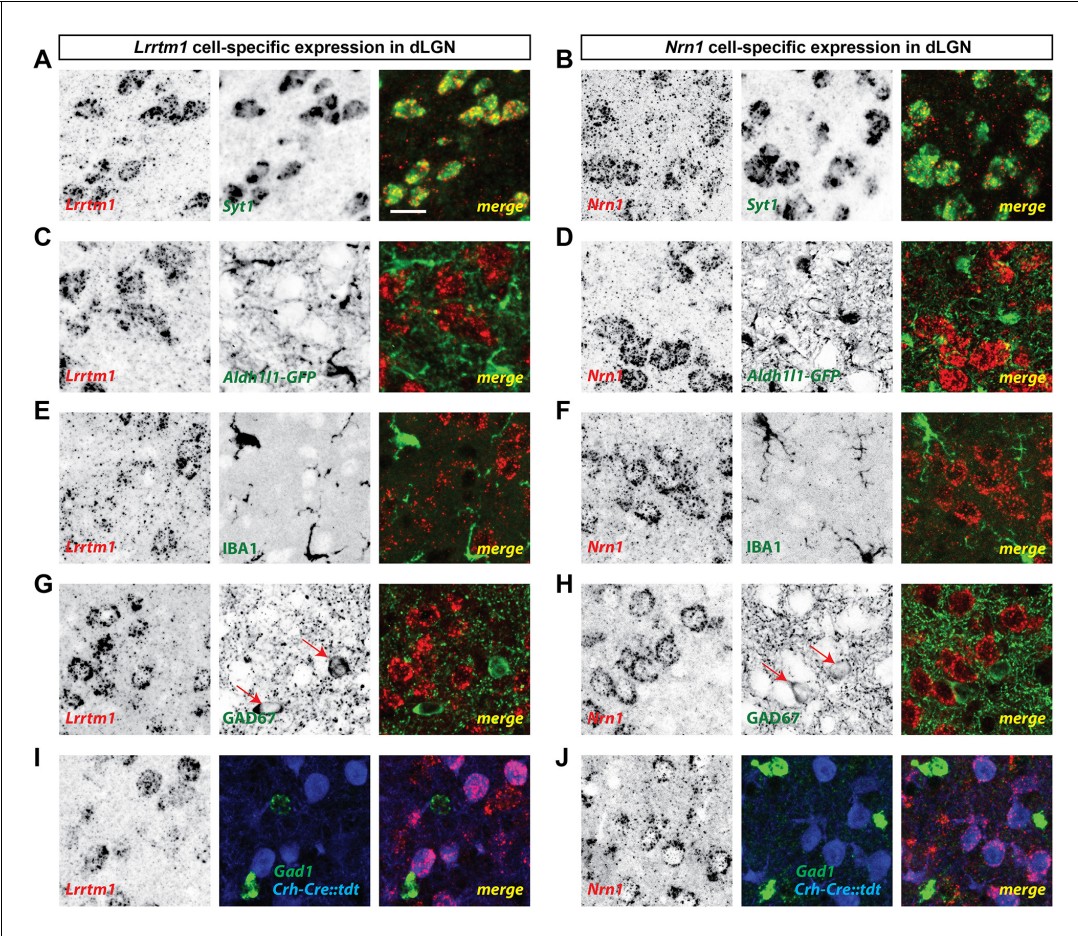

**Figure 4.** dLGN relay cells generate *Lrrtm1* and *Nrn1*. (A and B) Double in situ hybridization (ISH) for *Syt1* and either *Lrrtm1* (A) or *Nrn1* (B) in P14 wild type dLGN. (C and D) ISH for either *Lrrtm1* (C) or *Nrn1* (D) in dLGN of P14 *Aldh1l1-GFP* mice revealed no astrocytic expression of these mRNAs. (E and F) ISH for *Lrrtm1* (E) or *Nrn1* (F) and immunostaining (IHC) for the microglia marker IBA1 in P14 wild type dLGN. (G and H) ISH for *Lrrtm1* (G) or *Nrn1* (H) and IHC for GAD67 in P14 (*Lrrtm1*) and P25 (*Nrn1*) wild type mice revealed no mRNA expression by inhibitory interneurons. Red arrows depict GAD67[+] interneurons. (I and J) Double ISH for either *Lrrtm1* (I) or *Nrn1* (J) and *Gad1* in P25 C*rh-Cre::tdt* dLGN revealed mRNA expression by relay cells. Scale bar, 20 μm (A–J).

DOI: https://doi.org/10.7554/eLife.33498.009

less than four distinct inputs (identified by their unique colors). In contrast, the majority (80%) of clusters in controls contained more than four distinct retinal terminals.

Thus, there was a significant loss of complex RG synapse in the absence of LRRTM1. In fact, these numbers underrepresent the loss of retinal convergence in mutants, since our criteria for defining a complex RG synapse requires the presence of just two distinct retinal inputs. Not only was there a significant loss of complex RG synapses in mutants, but those complex synapses that remained contained significantly fewer retinal terminals. In control dLGN about 86% of complex RG synapses contained between 4–14 retinal terminals, whereas the majority of complex RG synapses in *Lrrtm1*[−/−] mutants contained only 2 or 3 inputs (*Figure 6C*). While the reduced number of complex RG synapses (and retinal inputs at the few persisting complex RG synapses) might be caused by fewer retinal axons in mutants, we failed to observe a significant loss of RGC axons in the optic nerves of *Lrrtm1*[−/−] mice (*Figure 6—figure supplement 1*).

Surprisingly, we also observed an increase in individual retinal terminal size in both simple and complex RG synapses in *Lrrtm1*[−/−] mice (*Figure 6D*). This increase in terminal size in *Lrrtm1*[−/−] mutant dLGN was accompanied by a significant increase in active zone number compared to control RG synapses (*Figure 6E*). However, when we normalized active zone number to terminal size there was no difference in the density of active zones in control and mutant RG synapses (*Figure 6F*).

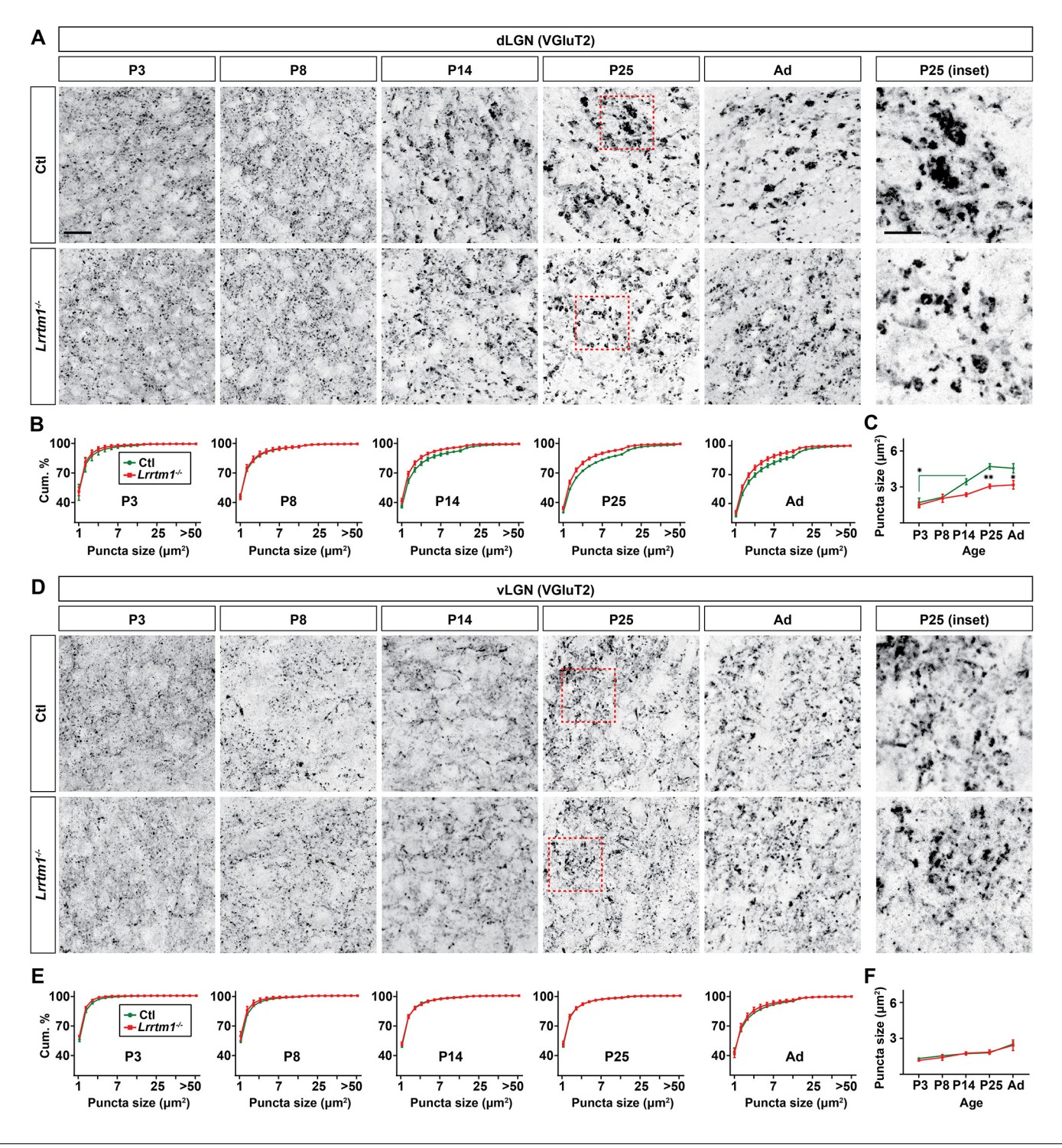

**Figure 5.** Loss of *Lrrtm1* results in smaller vglut2-positive puncta in dLGN but not vLGN. (**A and D**) Immunostaining of VGluT2$^+$ retinal terminals in dLGN (**A**) and vLGN (**D**) of littermate control and *Lrrtm1*$^{-/-}$ mice at P3, P8, P14 and P25 and adult mice (P60-85). Red boxes in P25 panels are shown in higher magnification on the right. (**B and E**) Cumulative (cum.) distribution of VGluT2$^+$ puncta size in control and *Lrrtm1*$^{-/-}$ mice dLGN (**B**) and vLGN (**E**). (**C and F**) Average VGluT2$^+$ puncta size in control and *Lrrtm1*$^{-/-}$ mice dLGN (**C**) and vLGN (**F**). Data represent Mean ±SEM; **p=0.0005, *p<0.05 by ANOVA. Scale bar, 20 µm (**A and C**), 10 µm (insets).

DOI: https://doi.org/10.7554/eLife.33498.010

The following figure supplements are available for figure 5:

*Figure 5 continued on next page*

*Figure 5 continued*

**Figure supplement 1.** Lack of *Lrrtm1* affects retinogeniculate synapses but not other synapses in dLGN.
DOI: https://doi.org/10.7554/eLife.33498.011
**Figure supplement 2.** Lack of *Nrn1* did not alter retinal nerve terminal development in dLGN.
DOI: https://doi.org/10.7554/eLife.33498.012

## Impaired visual behaviors in mice lacking LRRTM1

The functional consequence of LRRTM1 deletion and the loss of complex RG synapses was assessed by a two-alternative forced swim test (*Prusky et al., 2000*; *Wong and Brown, 2006*; *Huberman and Niell, 2011*). In this test, mice learn to associate a visual cue with a hidden platform that allows them to escape the water (*Figure 7A*). In order to confirm the necessity of vision for performing this task we asked whether $Atoh7^{-/-}$ mice (also called $Math5^{-/-}$), which are genetically blind (*Wang et al., 2001*), can detect the positive visual cue and find the hidden platform. $Atoh7^{-/-}$ mice were unable to perform this task, demonstrating the importance of vision in this assay (*Figure 7—figure supplement 1*).

To explore the role of LRRTM1 (and complex RG synapses) in vision, mice were trained for 8 days to detect a vertical grating (0.17 cycle per degree, cpd) on S+ monitor positioned above the hidden platform, compared with a gray screen or a horizontal grating display on the S- monitor (*Figure 7A*). Mice that exceeded 70% accuracy in locating the hidden platform were considered capable of detecting and discriminating the visual cues (*Prusky et al., 2000*). $Lrrtm1^{-/-}$ mutants and controls displayed equal abilities to distinguish the vertical gratings (i.e. the positive visual cue) from a gray screen or horizontal gratings at the end of training, although the initial learning phase of $Lrrtm1^{-/-}$ mice was modestly impaired (*Figure 7B,C*), which is in agreement with previous findings showing a delayed response of $Lrrtm1^{-/-}$ mutant mice to new environment (*Takashima et al., 2011*). Although we used vertical gratings as the positive cue in all subsequent experiments, we tested whether controls or mutants could also learn these tasks if horizontal gratings were the positive cue. We found no difference in mutants or controls learning to associate the hidden platform under a screen with horizontal gratings (*Figure 7—figure supplement 1B*). Moreover, in order to demonstrate that control or mutant mice were not capable of detecting the hidden platform itself (instead of associating it to the visual cue), we trained control and mutant mice (for 8 days) to associate the positive visual cue (S+) (with near 100% accuracy) with the platform. We then moved the platform below the negative visual cue (S-) and tested each mouse for 10 trials in day 9. Mutants and controls swam toward the positive visual cue that lacked the rescue platform, confirming they could not visually detect the hidden platform (*Figure 7—figure supplement 1C*).

By changing the frequency of the vertical bars, we next tested visual acuity in $Lrrtm1^{-/-}$ mutants. Results indicate that acuity was similar between mutants and controls, both falling below the 70% correct criteria at spatial frequencies above 0.57 cpd (*Figure 7D*). There was a statistically significant difference between $Lrrtm1^{-/-}$ and control mice at a single spatial frequency (0.62 cpd), however, at this frequency both performed poorly in the task (*Figure 7D*). Next, we altered the contrast of the vertical grating bars rather than the spatial frequency or orientation. Similarly, $Lrrtm1^{-/-}$ mutants failed to show significant differences compared to wild type mice (*Figure 7E*). Taken together these results suggest $Lrrtm1^{-/-}$ mutants do not exhibit deficits in visual acuity, simple pattern recognition or contrast sensitivity.

Next, we exposed mice to more complex visual tasks in which multiple features of the visual scene were altered at once. For example, we challenged mice to differentiate vertical and horizontal gratings while increasing the spatial frequency. In this more complex task, we found $Lrrtm1^{-/-}$ mutants performed significantly worse than controls (*Figure 7—figure supplement 1D*). Since there are conflicting data indicating spatial memory deficits in mice lacking LRRTM1 (*Voikar et al., 2013*; *Takashima et al., 2011*), we next adjusted our experimental design to rule out the influence of spatial memory impairment on performing the visual tasks. After 8 days of training with a new cohort of mutant and control mice, pattern discrimination was again tested while also increasing the spatial frequency or decreasing the contrast of both vertical and horizontal gratings. After each day of testing, we checked the ability of mice to perform the initial, standard discrimination task (i.e. to discriminating vertical and horizontal grating with 0.17 cpd and 100% contrast). Throughout the

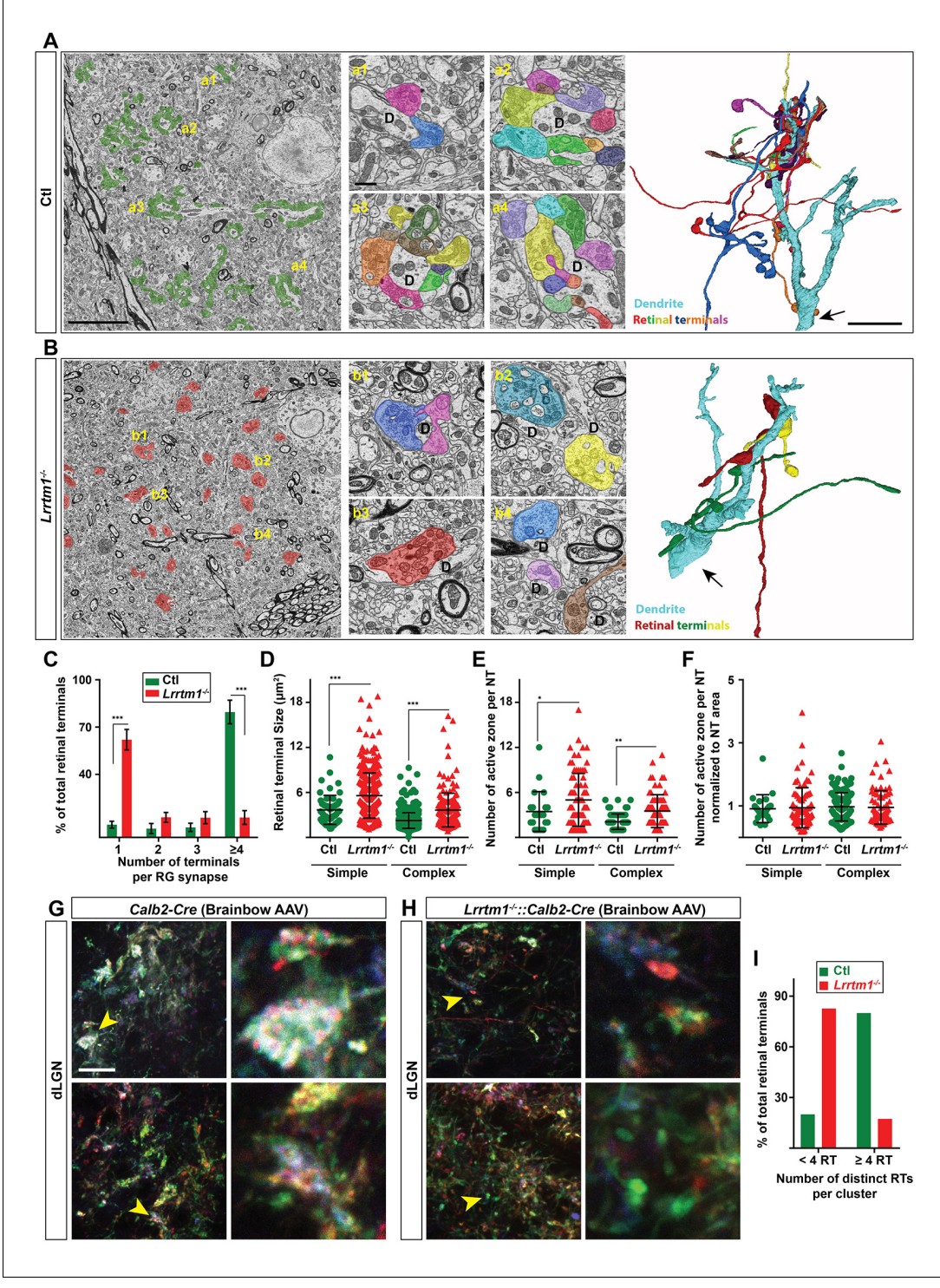

**Figure 6.** Loss of complex RG synapses in *Lrrtm1*$^{-/-}$ mice. (**A and B**) SBFSEM images of retinal terminals in P42 control (A, labeled in green) and *Lrrtm1*$^{-/-}$ (B, labeled in red) dLGN. RG synapses are depicted in insets a1-a4 (Ctl) and b1-b4 (*Lrrtm1*$^{-/-}$). In insets, each retinal terminal is depicted in a unique color. Similar colors in different insets do not represent axonal branches of the same RGC. 3D reconstruction of retinal terminals and relay cell dendrite are depicted on the right. The black arrows denote the position of dendrites stemming from relay cells somas. (**C**) Percentage of retinal terminals contributing to forming RG synapses with 1, 2, 3 or ≥4 distinct retinal terminals in P42 *Lrrtm1*$^{-/-}$ and control dLGN. Data represent Mean ±SEM; ***p<0.0001, by ANOVA. (**D–F**) Quantification of terminal size (**D**), actual number of active zones per terminal (**E**) and the active zones: terminal area ratio for simple and complex RG synapses in dLGN of *Lrrtm1*$^{-/-}$ and control mice (NT = nerve terminal). Data represent

*Figure 6 continued on next page*

*Figure 6 continued*

Mean ±SEM; ***p<0.0001, by ANOVA. (**G and H**) Retinal terminals were multicolor-labeled by injecting 1–2 µl brainbow AAVs into the vitreous humor of *Lrrtm1⁻/⁻::Calb2-Cre* and control mice. (**I**) Color analysis of clustered retinal terminals in wild type and *Lrrtm1⁻/⁻* mutants revealed a lower level of complex RG synapses in mutants. Scale bar, 10 µm (A and the 3D image), 20 µm (**G**), 1 µm (insets).

DOI: https://doi.org/10.7554/eLife.33498.013

The following figure supplement is available for figure 6:

**Figure supplement 1.** Loss of LRRTM1 resulted in a significant decrease in the number of complex RG synapses.

DOI: https://doi.org/10.7554/eLife.33498.014

---

experiments, *Lrrtm1⁻/⁻* mice failed to show any signs of memory deficits in this task. Interestingly, while control mice were able to discriminate vertical and horizontal bars at a spatial frequency of 0.32 cpd, *Lrrtm1⁻/⁻* mice performance dropped significantly under the 70% threshold during these more complex tasks (*Figure 7F*). Similarly, at lower contrast (i.e. 25% and 10%) mutant mice lacked the sensitivity to discriminate vertical and horizontal grating patterns (*Figure 7G*). As another set of controls, we repeated these behavioral tasks with *Nrn1⁻/⁻* mutants and found no deficit in their performance in either the simple or complex visual tasks (*Figure 7—figure supplement 1E–J*). Taken together, these results indicate that mice lacking LRRTM1 and complex RG synapses exhibit abnormalities in performing more complex visual tasks.

## Disscussion

The recent discovery by three independent groups (*Hammer et al., 2015*; *Rompani et al., 2017*; *Morgan et al., 2016*) that a shockingly high level of retinal convergence exists onto thalamic relay cells in rodents has raised a number of questions for the field. *Are these converging inputs functional? Is retinal convergence in visual thalamus important for vision? What are the developmental mechanisms that underlie the establishment of retinal convergence in dLGN?* We are only beginning to scratch the surface in answering these questions. For example, *Litvina and Chen (2017)* only recently applied an optogenetic approach to demonstrate a higher level of functional retinal convergence onto dLGN relay cells. Here, we sought to address the last question mentioned above, what mechanisms underlie the establishment of retinal convergence in visual thalamus. Using an unbiased screen, we identified LRRTM1 as a target-derived cue necessary for the formation of retinal convergence onto dLGN relay cells. Analysis in LRRTM1-deficient mice revealed that the lack of this synaptic adhesion molecule led to impaired visual function. We interpret these results to suggest that complex RG synapses are necessary for visual processing. It is important to point out that we also observed an increase in the retinal never terminal size (and active zone number) in the absence of LRRTM1 and this may also contribute to deficits in visual behaviors.

### LRRTM1 as a target-derived synaptic organizer in visual thalamus

LRRTMs are transmembrane proteins that act as transsynaptic signals to trigger excitatory synaptogenesis (*Linhoff et al., 2009*; *de Wit et al., 2009*; *de Wit and Ghosh, 2014*; *Um et al., 2016*). When present in the postsynaptic membrane, LRRTM1 binds to the extracellular domain of neurexins to induce presynaptic differentiation in contacting axons (*Linhoff et al., 2009*; *Siddiqui et al., 2010*). In visual thalamus, LRRTM1 is specifically expressed by relay cells (and not other cells) and its transsynaptic partners, neurexins, are generated by RGCs (*Figure 2—figure supplement 1*) (*Sajgo et al., 2017*; *Shigeoka et al., 2016*). Therefore, based on our results, we hypothesized that LRRTM1-neurexin interactions are critical for the formation of complex RG synapses. Although the necessity of neurexins in retinogeniculate connectivity has yet to be thoroughly examined, the loss of CASK, a MAGUK protein necessary for trafficking neurexins to the presynaptic membrane, leads to abnormal retinogeniculate connectivity and optic nerve hypoplasia (*LaConte et al., 2016*; *Srivastava et al., 2016*; *Moog et al., 2011*; *Liang et al., 2017*).

It is important to point out that neurexins have other postsynaptic partners expressed in visual thalamus, including neuroligins and other LRRTMs (*Figure 2—figure supplement 1B–G*) (*Laurén et al., 2003*; *Varoqueaux et al., 2006*), each capable of inducing excitatory synaptogenesis elsewhere in the brain or *in vitro* (*Fox and Umemori, 2006*; *Craig et al., 2006*; *Ko et al., 2009*;

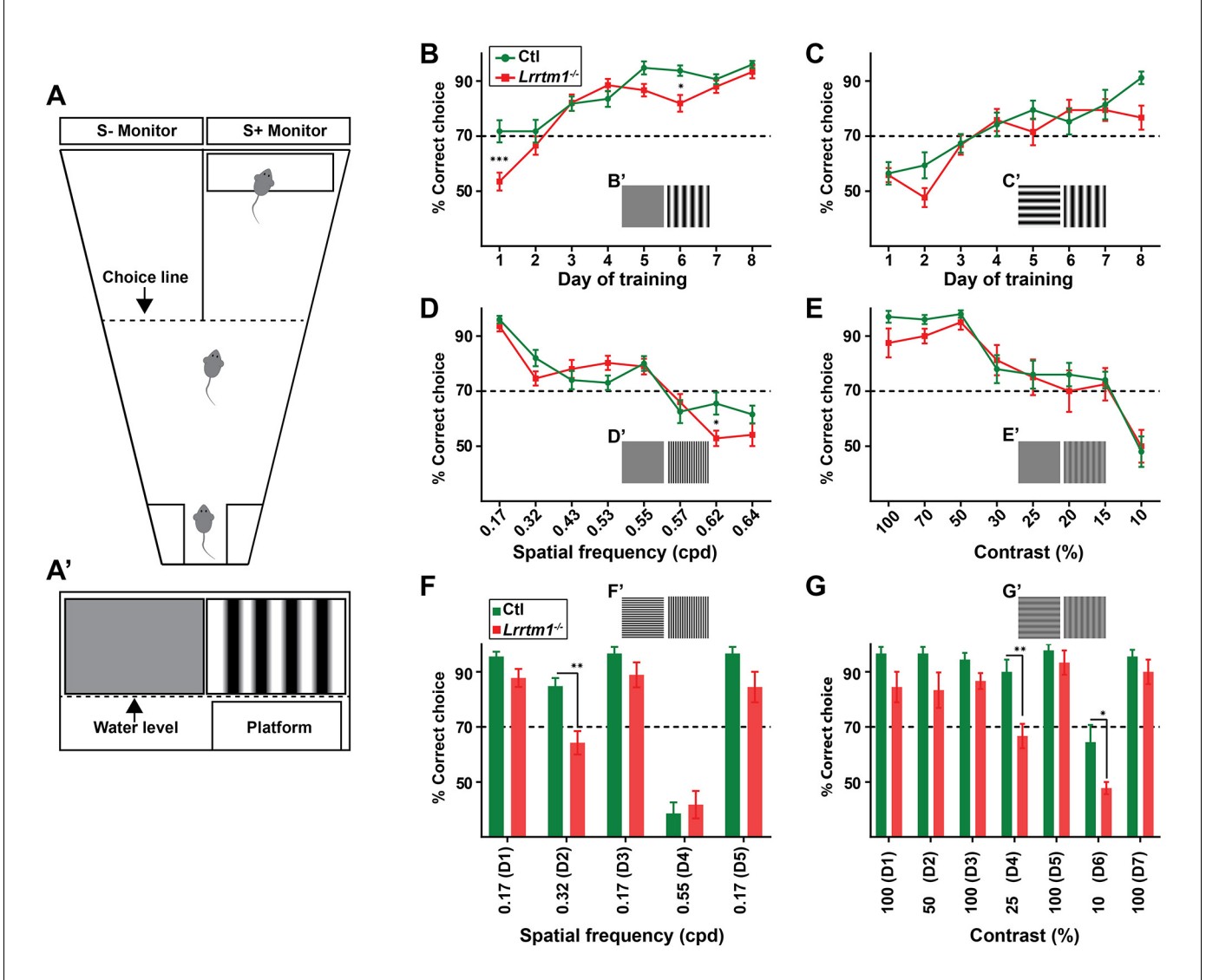

**Figure 7.** Complex RG synapses are required for visual behaviors. (A) Schematic diagrams depicting the two-alternative forced swim behavior task. A' depicts a mouse's view of the visual displays (e.g. vertical gratings). (B and C) Training $Lrrtm1^{-/-}$ (red) and control (green) mice to detect a vertical grating display versus a gray screen (B, n = 20 for Ctl; and 21 for $Lrrtm1^{-/-}$) or horizontal grating display (C, n = 19 for Ctl; and 18 for $Lrrtm1^{-/-}$). Examples of visual displays are depicted in B' and C'. (D and E) Percentage of correct choices made by $Lrrtm1^{-/-}$ (red) and control (green) mice for detection of vertical gratings with increasing spatial frequency (D; cpd, cycle per degree, n = 20 for Ctl; and 21 for $Lrrtm1^{-/-}$) or decreasing contrast versus a gray screen (E, n = 10 for Ctl; and nine for $Lrrtm1^{-/-}$). Examples of visual displays are depicted in D' and E'. (F and G) Correct choices made by $Lrrtm1^{-/-}$ (red) and control (green) mice for discriminating vertical grating from horizontal grating either with increasing spatial frequency (F, n = 10 for Ctl; and 10 for $Lrrtm1^{-/-}$) or with decreasing contrast (G, n = 10 for Ctl; and nine for $Lrrtm1^{-/-}$). D1-D7 are the consecutive days of the test phase. Examples of visual displays are depicted in F' and G'. For B-G, dashed line represents the 70% correct threshold for successful completion of task. All data are shown as Mean ±SEM; ***p<0.0001, **p<0.01, *p<0.05 by ANOVA.

DOI: https://doi.org/10.7554/eLife.33498.015

The following figure supplements are available for figure 7:

**Figure supplement 1.** Visual behaviors in $Lrrtm1^{-/-}$, $Nrn1^{-/-}$ and $Atoh7^{-/-}$ mice.

DOI: https://doi.org/10.7554/eLife.33498.016

**Figure supplement 2.** Loss of $Lrrtm1^{-/-}$ did not alter the distribution of VGluT1[+] and VGluT2[+] terminals in visual cortex.

DOI: https://doi.org/10.7554/eLife.33498.017

*Siddiqui et al., 2010*). The presence of LRRTM1, other LRRTMs, and neuroligins in dLGN raises an interesting possibility that simple and complex RG synapses may be assembled through different postsynaptic interactions with neurexins. As such, astrocytes may also contribute to the signals that regulate simple or complex RG synapses. Retinal terminals in simple RG synapse are ensheathed by astrocytic processes (*Hammer et al., 2014*; *Bickford et al., 2010*), and these astrocytes are known to produce extracellular factors capable of bridging neurexin–neuroligin interactions to facilitate excitatory synaptogenesis (*Kucukdereli et al., 2011*; *Singh et al., 2016*).

The presence of multiple postsynaptic neurexins partners in dLGN also raises the possibility that their abundance (or overabundance) may prevent some level of activity-dependent refinement in dLGN. Overexpression of different combinations of neurexin-binding partners in postsynaptic neurons has been shown to impair synapse elimination in vitro (*Ko et al., 2011*). The emergence of complex RG synapses at eye-opening may therefore represent synapses with an overabundance of neurexin-binding receptors, in which strong trans-synaptic adhesion prevent complete activity-dependent RG refinement. While certainly possible, we see this as unlikely given the dramatic refinement of retinal arbors around eye-opening in mice, and since this RG refinement itself gives rise to retinal bouton clustering (*Dhande et al., 2011*; *Hong et al., 2014*).

### Retinal convergence: artifact or by design?

The discovery of an extraordinary level of retinal convergence on mouse relay cells has left the field pondering whether such convergence is an artifact of impaired refinement (as described above) or whether there is functional significance to such 'fuzzy' connectivity (as one group has termed this retinogeniculate convergence) (*Morgan et al., 2016*). It is easy to discount the importance of retinal convergence onto relay cells and the role that complex RG synapses may play in vision, since many groups (including our own [*Hammer et al., 2014*]) have demonstrated that relay cells receive a very small number of strong, functional inputs from the retina (*Chen and Regehr, 2000*; *Jaubert-Miazza et al., 2005*; *Hooks and Chen, 2006*; *Litvina and Chen, 2017*). Many of the techniques used to identify high levels of retinogeniculate convergence in mice have been anatomical in nature (e.g. ultrastructural analysis, anterograde multicolor labeling of RGCs and retrograde trans-synaptic tracing) (*Hammer et al., 2015*; *Morgan et al., 2016*; *Rompani et al., 2017*), leading to the possibility that 'form' does not fit 'function' in mouse visual thalamus. Recent optogenetic analysis of the RG circuit in mice has revealed a substantially higher level of functional retinal convergence on relay cells, however the strength of these inputs widely varies (*Litvina and Chen, 2017*). Functional roles for weak RG synapses remain unclear.

In the present study, we took advantage of the loss of complex RG synapses in *Lrrtm1*$^{-/-}$ mice to begin to shed light on the functional significance of retinal convergence on thalamic relay cells. While the ability of *Lrrtm1*$^{-/-}$ mice to perform tasks with simple visual cues appeared unaltered compared with controls, they performed poorly on tasks where more than one feature of the visual scenes was altered at once. Although these mice lack LRRTM1 globally, such deficits are likely the direct result of impaired RG circuitry for several reasons. First, LRRTM1 is largely absent from retina and visual cortex (*Figure 3E–F*), sites whose function are required for the performance of these visual tasks. Second, global deletion of LRRTM1 failed to result in synaptic or cytoarchitectural changes in other brain regions that process visual information (*Figure 7—figure supplement 2A–F*). For these reasons, we believe that results presented here provide the first clues that complex RG synapses (and retinal convergence) are not functionally insignificant artifacts of impaired or incomplete activity-dependent refinement, but rather are an important component of processing and relaying visual information from the retina to visual cortex.

## Materials and methods

**Key resources table**

| Reagent type (species) or resource | Designation | Source or reference | Identifiers |
|---|---|---|---|
| Virus, AAV9 | *AAV9.hEF1a.lox.TagBFP.lox.eYFP.lox. WPRE.hGH-InvBYF* | Penn Vector Core | AV-9-PV2453 |

*Continued on next page*

*Continued*

| Reagent type (species) or resource | Designation | Source or reference | Identifiers |
|---|---|---|---|
| Virus, AAV9 | *AAV9.hEF1a.lox.mCherry.lox.mTFP1. lox.WPRE.hGH-lnvCheTF* | Penn Vector Core | AV-9-PV2454 |
| Mouse (C57/Bl) | *Lrrtm1$^{-/-}$ (Lrrtm1$^{tm1Lex}$)* | MMRRC | RRID:MMRRC_031619-UCD |
| Mouse (C57/Bl) | *Atoh7$^{tm1Gla}$ (Atoh7$^{-/-}$)* | MMRRC | RRID:MMRRC_042298-UCD |
| Mouse (C57/Bl) | *Aldh1l1-EGFP* | MMRRC | RRID:MMRRC_011015-UCD |
| Mouse (C57/Bl) | *Nrn1$^{-/-}$ (Nrn1$^{tm1.2Ndiv}$)* | The Jackson Laboratory | RRID:IMSR_JAX:018402 |
| Mouse (C57/Bl) | *Rosa-stop-tdT* | The Jackson Laboratory | RRID:IMSR_JAX:007905 |
| Mouse (C57/Bl) | *Crh-Cre* | MMRRC | RRID:MMRRC_030850-UCD |
| Mouse (C57/Bl) | *Calb2-Cre* | The Jackson Laboratory | 10774 |
| antibody | mouse anti-GAD67 | EMD Millipore | MAB5406, RRID:AB_2278725 |
| antibody | mouse anti-actin | EMD Millipore | MAB1501, RRID:AB_2223041 |
| antibody | rabbit anti-IBA1 | Wako | 019–19741, RRID:AB_839504 |
| antibody | rabbit anti-mGluR1a | Frontier Institute co.,ltd | RRID:AB_2571799 |
| antibody | rabbit anti-VGluT1 | Synaptic Systems | 135402, RRID:AB_2187539 |
| antibody | rabbit anti-VGluT2 | Synaptic Systems | 135511 |
| antibody | sheep anti-LRRTM1 | Synaptic Systems | AF4897, RRID:AB_10643427 |
| antibody | sheep (POD)-conjugated anti-DIG | Roche | 11426346910 |
| antibody | sheep (POD)-conjugated anti-FL | Roche | 11207733910 |
| commercial assay or kit | PrepX PolyA mRNA Isolation Kit | Wafergen | 400047 |
| commercial assay or kit | PrepX RNA-Seq for Illumina Library Kit, 48 samples | Wafergen | 400046 |
| commercial assay or kit | Quant-iT dsDNA HS Kit | Invitrogen | Q33120 |
| commercial assay or kit | Superscript II Reverse Transcriptase First Strand cDNA Synthesis kit | Invitrogen | 18064014 |
| commercial assay or kit | Aurum Total RNA Fatty and Fibrous Tissue kit | BioRad | 7326870 |
| commercial assay or kit | pGEM-T Easy Vector Systems | Promega | A1360 |
| commercial assay or kit | Ambion MAXIscript T7 In Vitro Transcription Kit | Thermo Fisher Scientific | AM1312 |
| commercial assay or kit | | | |
| commercial assay or kit | iTaq SYBRGreen Supermix | BioRad | 1725124 |
| commercial assay or kit | Tyramide Signal Amplification (TSA) systems | PerkinElmer | NEL75300 1KT |
| commercial assay or kit | Amersham ECL Prime Western Blotting Detection Reagent | GE Healthcare Life Sciences | RPN2236 |
| commercial assay or kit | TruSeq PE Cluster Kit v3-cBOT-HS | illumina | PE-401–3001 |
| commercial assay or kit | TruSeq SBS Kit v3-HS (200-cycles) | illumina | FC-401–3001 |
| chemical compound, drug | Alexa-conjugated cholera toxin beta subunit | Thermo Fisher Scientific | C22841 |
| chemical compound, drug | Fluorescein RNA Labeling Mix | Roche | 11685619910 |
| chemical compound, drug | DIG RNA Labeling Mix | Roche | 11277073910 |
| chemical compound, drug | Proteinase K | Thermo Fisher Scientific | EO0491 |
| chemical compound, drug | Paraformaldehyde, EM grade | EMS | 19202 |

*Continued on next page*

*Continued*

| Reagent type (species) or resource | Designation | Source or reference | Identifiers |
|---|---|---|---|
| chemical compound, drug | Sodium cadodylate | EMS | 12300 |
| chemical compound, drug | Tissue Freezing Medium | EMS | 72592 |
| chemical compound, drug | Glutaraldehyde | EMS | 16220 |
| chemical compound, drug | Prehybridization Solution | Sigma | P1415 |
| chemical compound, drug | Heparin Sodium | Fisher Scientific | BP2425 |
| chemical compound, drug | Yeast RNA | Roche | 10109223001 |
| chemical compound, drug | Blocking reagent (ISH) | Roche | 11096176001 |
| software, algorithm | TrakEM2 | ImageJ plugin | RRID:SCR_008954 |
| software, algorithm | Fiji | Fiji | RRID:SCR_002285 |
| software, algorithm | ImageJ | NIH | RRID: SCR_003070 |
| software, algorithm | Prism | GraphPad | RRID: SCR_002798 |
| software, algorithm | Gabor-patch generator | https://www.cogsci.nl/ | N/A |
| sequence-based reagent | Cloning primers for *Gad1*: F:TGTGCCCAAACTGGTCCT; R:TGGCCGATGATTCTGGTT | Integrated DNA Technologies | N/A |
| sequence-based reagent | qPCR primer for *Nrn1*: F:TTCCCCCGCGTTCTCTAAAC; R:GCCTGCACCAGGTAAGCTAT | Integrated DNA Technologies | N/A |
| sequence-based reagent | qPCR primer for *Lrrtm1*: F:AGCAGCTGAATGGAGGTTGTC; R:AGTGTAGACAGAGGCCGAGTAG | Integrated DNA Technologies | N/A |
| sequence-based reagent | qPCR primer for *Gapdh*: F:CGTCCCGTAGACAAA ATG GT; R:TTGATG GCAACAATC TCCAC | Integrated DNA Technologies | N/A |
| recombinant DNA reagent | *Nrn1* (5367281) | Dharmacon | MMM1013-202769896 |
| recombinant DNA reagent | *Lrrtm1* (5321979) | Dharmacon | MMM1013-202769075 |
| recombinant DNA reagent | *Syt1* (5363062) | Dharmacon | MMM1013-202709704 |

## Animals

CD1 and C57/BL6 mice were obtained from Charles River (Wilmington, MA) or Harlan (Indianapolis, IN). *Lrrtm1$^{-/-}$* mice were obtained from MMRRC (stock # 031619-UCD), *Nrn1$^{-/-}$* (stock # 018402), *Calb2-Cre* (stock # 010774) and Rosa-stop-tdT mice (stock # 007905) were all obtained from Jackson Laboratory. *Crh-Cre* (stock # 030850-UCD) and *Aldh1l1-EGFP* (stock # 011015-UCD) mice were obtained from W. Guido (University of Louisville) and S. Robel (Virginia Tech), respectively. *Atoh7$^{-/-}$* (stock# 042298-UCD) were obtained from S. W. Wang and were described previously (*Wang et al., 2001*). Mice were housed in a 12 hr dark/light cycle and had *ad libitum* access to food and water. Late dark-reared (LDR) mice were placed in a light-tight room from P20-P31. Dissections of LDR mice were performed at P31 in a dark room under red light. All experiments were performed in compliance with National Institutes of Health (NIH) guidelines and protocols and were approved by the Institutional Animal Care and Use Committee (IACUC# 15-137VTCRI, 15-167VTCR and 15-174VTCRI) and Institutional Biosafety Committee (IBC# 15–038) at Virginia Tech.

## Immunohistochemistry (IHC)

Anesthetized mice were transcardially perfused with phosphate-buffered saline (PBS; pH 7.4) and 4% paraformaldehyde in PBS (PFA; pH 7.4). Dissected brains and eyes were post-fixed in 4% PFA for 12–16 hr at 4°C. Tissues were cryopreserved in 30% sucrose solution for 2–3 days, embedded in Tissue Freezing Medium (Electron Microscopy Sciences, Hatfield, PA), and cryosectioned (16 µm sections). Sections were air-dried onto Superfrost Plus slides (Fisher Scientific, Pittsburgh, PA) and frozen at −80°C until further processing. For IHC, slides were incubated in blocking buffer (2.5% bovine serum albumin, 5% Normal Goat Serum, 0.1% Triton-X in PBS) for 1 hr. Primary antibodies were

diluted in blocking buffer as follows: GAD67 (Millipore MAB5406) 1:700; IBA1 (Wako 019–19741) 1:1000; VGluT2 (Synaptic Systems 135511) 1:500; VGluT1 (Synaptic Systems 135402) 1:700; mGluR1a (Frontier Institute co. AB_2571799) 1:250 and incubated on tissue sections for >12 hr at 4°C. After washing three times in PBS, fluorescently conjugated secondary antibodies (1:1000 in blocking buffer) were incubated on sections for 1 hr at room temperature. After five washes with PBS, sections were stained with DAPI (1:5000 in water) and were mounted with Vectashield (Vector Laboratories, Burlingame, CA). Images were acquired on a Zeiss LSM 700 confocal microscope. When comparing sections from different age groups or genotypes, images were acquired with identical parameters. A minimum of three animals (per genotype and per age) were compared in all IHC experiments.

## Riboprobe production

pCMV-SPORT6 Plasmids carrying *Syt1* (cat # 5363062), *Nrn1* (cat # 5367281), and *Lrrtm1* (cat # 5321979) were obtained from GE Dharmacon. *Gad1* 1 Kb cDNA (corresponding to nucleotides 1099–2081) was generated using Superscript II Reverse Transcriptase First Strand cDNA Synthesis kit (cat # 18064014, Invitrogen, La Jolla, CA) according to the manufacturer manual, amplified by PCR using primers mentioned in the primers list, gel purified, and then cloned into a pGEM-T Easy Vector using pGEM-T Easy Vector kit, (cat # A1360, Promega, Madison, WI) according to the kit manual. Sense and anti-sense riboprobes against *Gad1, Syt1, Nrn1*, and *Lrrtm1* were synthesized from 5 µg linearized plasmids using digoxigenin-(DIG) or fluorescein-labeled uridylyltransferase (UTP) (cat # 11685619910, cat # 11277073910, Roche, Mannheim, Germany) and the MAXIscript in vitro Transcription Kit (cat # AM1312, Ambion, Austin, TX) according to the kit manual. 5 µg of Riboprobes (20 µl) were hydrolyzed into ~0.5 kb fragments by adding 80 µl of water, 4 µl of NaHCO3 (1 M), 6 µl Na2CO3 (1 M) and incubating the mixture in 60°C for specific amounts of time determined for each probe by the following formula: Time=$(X_{kb}-0.5)/(X_{kb}*0.055)$, where X is the full length of the RNA probe. RNA fragments were finally precipitated in 250 µl 100% ethanol containing 5 µl Acetic acid, 10 µl NaCl (5 M) and 1 µl glycogen (5 mg/ml). Finally, the pellet dissolved in 50 µl of RNAase-free water.

## In situ hybridization (ISH)

ISH was performed on 16 µm sections prepared as described above. Sections were fixed in 4% PFA for 10 min, washed with PBS for 15 min, incubated in proteinase K solution (1 µg/ml in 50 mM Tris PH 7.5, 5 mM EDTA) for 10 min, washed with PBS for 5 min, incubated in 4% PFA for 5 min, washed with PBS for 15 min, incubated in acetylation solution (196.6 ml water, 2.6 ml triethanolamin, 0.35 ml HCl, 0.5 ml acetic acid) for 10 min, washed with PBS for 10 min, incubated in 0.1% triton (in PBS) for 30 min, washed with PBS for 40 min, incubated in 0.3% $H_2O_2$ (in water) for 30 min, washed with PBS for 10 min, pre-hybridized with hybridization solution (50 ml of Sigma 2X prehyb solution, 25 mg Roche yeast RNA and 8 mg heparin) for 1 hr, hybridized with 50 µl of heat-denatured diluted riboprobes (1–2 µl of riboprobe in 50 µl hybridization solution heated for 10 min in 70°C), mounted with cover slips and kept at 60°C overnight. On day 2, coverslips were gently removed in 60°C preheated 2X saline-sodium citrate (SSC) buffer, and slides were washed 5 times in 60°C preheated 0.2X SSC buffer for 2–3 hr at 60°C. Slides were washed three times with Tris-buffered saline (TBS) and blocked for 1 hr with blocking buffer (0.2% Roche blocking reagent, 10% lamb serum in TBS) prior to overnight 4°C incubation with horseradish peroxidase (POD)-conjugated anti-DIG or anti-fluorescent antibodies (cat # 11426346910 and cat # 11207733910, Roche). On day 3, bound riboprobes were detected by staining with Tyramide Signal Amplification (TSA) system (cat # NEL75300 1KT, PerkinElmer, Shelton, CT). For double ISH, sections were washed in TBS after the TSA reaction, then incubated in 0.3% $H_2O_2$ for 30 min, washed with TBS for 10 min, incubated with the second POD-conjugated antibody in blocking buffer and detected with TSA system as described above. Images were obtained on a Zeiss LSM700 confocal microscope. A minimum of three animals per genotype and age were compared in ISH experiments.

## Quantitative real time PCR (qPCR)

Pooled tissues (5–7 animals per sample) were isolated from P3, P8, P12 and P25 mice, and RNA was purified using the Aurum Total RNA Fatty and Fibrous Tissue kit (cat # 7326870, BioRad) according

to the kit manual. cDNAs were generated with Superscript II RT (Invitrogen). qPCR was performed on a CFX Connect real time system (BioRad) using iTaq SYBRGreen Supermix (cat # 1725124, Bio-Rad) according to the kit protocol. The following cycling conditions were used with 12.5 ng of cDNA: 95°C for 30 s and 42 cycles of amplification (95°C for 10 s, 60°C for 30 s) followed by a melting curve analysis. Relative quantities of RNA were determined using the ΔΔ-CT method (*Schmittgen and Livak, 2008*). A minimum of n = 3 biological replicates (each in triplicate) was run for each gene. Each individual run included separate Glyceraldehyde-3-Phosphate Dehydrogenase (*gapdh*), Actin, or 18 s rRNA control reactions. qPCR primers can be found in the primer list.

## Western blot

Mice were perfused with PBS, brains removed, and d- and vLGN were dissected separately in ice-cold PBS. Tissues were pooled from >5 littermates per group and subsequenctly lysed in modified loading buffer containing 50 mM Tris–HCl (pH 6.8), 2% sodium dodecyl sulfate (SDS), 10% glycerol, and protease inhibitors (1 mM PMSF). Samples were homogenized, boiled for 10 min, and insoluble material was removed. Protein concentrations were determined by Micro BCA Protein Assay Kit (cat # 23235, Pierce, Rockford, IL). Equal amounts of protein were loaded and separated by SDS-PAGE and transferred to a PVDF membrane as described previously (*Fox et al., 2007*). After blocking in 5% non-fat milk in PBS (containing 0.05% Tween), PVDF membranes were incubated with primary antibodies (LRRTM1 [Synaptic Systems AF4897], Actin [EMD Millipore MAB1501]), followed by HRP-conjugated secondary antibodies. Immunoblotted proteins were detected with Amersham ECL Prime Western Blotting Detection Reagent (cat # RPN2236).

## Intraocular injection of anterograde tracers and AAVs

For intraocular injections, mice were anesthetized with isoflurane or hypothermia, and 1–2 μl of 1 mg/ml CTB was injected into the eye intravitreally with a fine glass pipette attached to a picospritzer. After 2 days, perfused and PFA fixed brains were sectioned (90 μm) using a Vibratome (HM650v, ThermoFisher). Sections were stained with DAPI and mounted with Vectashield (Vector Laboratories, Burlingame, CA). Images were acquired on a Zeiss LSM 700 confocal microscope. A similar approach was used to inject 1–2 μl of a 1:1 mixture of the following AAVs into the eyes: AAV9.hEF1a.lox.TagBFP.lox.eYFP.lox.WPRE.hGH-InvBYF (AV-9-PV2453, 3.47e13 gc/ml) and AAV9.hEF1a.lox.mCherry.lox.mTFP1.lox.WPRE.hGH-InvCheTF (AV-9-PV2454, 1.04e13 gc/ml). AAVs were injected into the eyes of P0 or P12 mice and 1–2 weeks after the injection, mice were anesthetized, perfused, and their brains were fixed in 4% PFA overnight. Brains were then sectioned (90 μm) using a Vibratome and sections were mounted with Vectashield. Images were acquired on a Zeiss LSM 700 confocal microscope.

## Serial block-face scanning electron microscopy

Mice were perfused with 0.1M sodium cacodylate buffer containing 4% PFA and 2.5% glutaraldehyde. Brains were immediately vibratomed (300 μm coronal sections), and dLGN tissues were dissected and shipped to Renovo Neural (Cleveland, OH). Processing and image acquisition were performed as described in detail previously (*Mukherjee et al., 2016*; *Hammer et al., 2014*). Serial image stacks were analyzed using TrakEM2 in Fiji (*Cardona et al., 2012*). Presence of synaptic vesicles and pale mitochondria have been used as features to distinguished retinal terminals from non-retinal terminals in dLGN (*Hammer et al., 2014*; *Bickford et al., 2010*). Analysis of data sets were performed independently by four researchers to ensure unbiased results.

## RNA sequencing

RNA was isolated from vLGN and dLGN at four different ages (P3, P8, P12 and P25) and was shipped to the Genomics Research Laboratory at Virginia Tech's Biocomplexity Institute for RNAseq analysis. Quality of total RNA was checked on Agilent BioAnalyzer 2100 (Agilent Technologies, Santa Clara CA). Libraries were generated using Apollo 324 Robot (Wafergen, CA). 500 ng of total RNA (with RIN ≥9.0) was enriched for polyA RNA using PrepX PolyA mRNA Isolation Kit (cat # 400047, Wafergen, Fremont, CA) and was then onverted into a library of template molecules using the PrepX RNA-Seq for Illumina Library Kit (cat # 400046, Wafergen, Fremont, CA). Validation of the 280–300 bp libraries (160–180 bp insert) was completed using an Agilent 2100 Bioanalyzer and quantitated

using Quant-iT dsDNA HS Kit (cat # Q33120, Invitrogen). Eight individually indexed cDNA libraries were pooled and sequenced on an Illumina HiSeq, resulting in a minimum of 40–50 million reads. Libraries were clustered onto a flow cell using Illumina's TruSeq PE Cluster Kit v3-cBOT-HS (cat # PE-401–3001), and sequenced 2 × 100 PE using TruSeq SBS Kit v3-HS (200-cycles) (cat # FC-401–3001). Low-quality base calls, sequences with low-complexity tails, and adaptor sequences were removed using a combination of Btrim and EA-utils. Sequencing reads were then aligned to the mouse genome (Tophat2/Bowtie) and expression determined via HTSeq counting. DESeq2 has been used to determine fold change and statistical significance of changes between samples.

## Visual behavior tasks

Two alternative forced swim tasks were performed in a trapezoid shaped pool (sides a = 25 cm, b = 80 cm, c and d = 143 cm) with two side-by-side monitors (19 inches, V196L, Acer) placed at the wide end (b) of the tank and separated by a black divider (42 cm). Detailed instructions for the apparatus were described previously (*Prusky et al., 2000*). A rescue platform (37 cm ×13 cm × 14 cm) was hidden under water below the monitor with the positive visual cue (termed the S+ side). Visual cues (i.e. different grating pattern) were generated in the Gabor-patch generator (https://www.cogsci.nl/gabor-generator). The visual cue and hidden platform were moved to the right or left screens in a pseudorandom manner with the following orders: LRLLRLRRLR, RLRRLRLLRL, RRLRLLRLRL and LLRLRRLRLR. During the behavioral tasks the room was dark, but a 60 W bulb was positioned above the holding cages. During the visual tasks, mice were held in separate cages which were placed on heating pads and lined with paper towels. A day before starting experiments, mice were acclimated to the experimenter and the pool through handling, a 1–2 min period of direct contact with the hidden platform at either arms, and submersion into the water at gradually-increasing distances from the hidden platform. The ability of mice to detect and associate the S+ monitor displaying vertical gratings with the rescue platform (in contrast to the lack of a platform beneath the S- screen that displayed either a gray or horizontal gratings) was assessed. Behavioral tasks included a training phase (8 days) and a testing phase (10–12 days). For training phases, mice were placed at the release chute and given one minute to find the platform for 8–10 trials per day. A trial was recorded as a correct choice if a mouse passed the choice line on the S+ side, while passing the choice line on the S- side was recorded as an incorrect choice. After arriving at the rescue platform, mice were placed back into their individual cages only if they made the correct choice. When a mouse made an incorrect choice, it was placed back at the release chute to perform another trial immediately before going back to its home cage. After 8 days of training, mice learned to find the positive visual cue (i.e. vertical gratings for most tasks, however horizontal gratings in *Figure 7—figure supplement 1B*) with a >80% accuracy. To test visual acuity and contrast sensitivity, we increased the spatial frequency and decreased the contrast of vertical gratings (i.e. the S+ cue), respectively. In the testing phase of the detection tasks, 10 trials of a given task (e.g. detection of vertical gratings with spatial frequency of 0.32 cpd versus a gray screen) were performed in 10 consecutive days (one per day). For the testing phase of the discrimination tasks two different approaches were used with different cohort of *Lrrtm1*$^{-/-}$, *Nrn1*$^{-/-}$ and control mice. First, a similar approach as the one mentioned above for the detection task (*Figure 7—figure supplement 1J*). Second, 10 trials of a task (e.g. discrimination of vertical gratings with spatial frequency of 0.32 cpd versus horizontal gratings with spatial frequency of 0.32 cpd) were tested in a single day (*Figure 7F, G*; *Figure 7—figure supplement 1I,J*). No more than six animals were tested in a given session. Each mouse (P56-90) performed no more than 10 trials per day.

## Quantifications and statistics

For quantifying the size of retinal terminals labeled with fluorescent CTB or VGluT2 immunostaining, the area of isolated puncta (which may contain one or more RGC terminals) were measured in 20X or 40X confocal images of dLGN and vLGN sections by semi-manual selection of the puncta in the ImageJ. 3–7 animals (three sections per animals) were analyzed per age and genotype and the cumulative frequencies of different terminal sizes were obtained. Two-way ANOVA analysis was used to determine any significant change in the distribution of retinal terminal sizes between groups. Intensity and density of the signals in immunostained images of dLGN, vLGN and vCTX were measured in ImageJ. 3–7 animals (three sections per animals) were analyzed per genotype and age and

the mean values were compared between groups. T-test or ANOVA were used to determine any significant difference of the mean values between groups.

For SBFSEM, retinal terminals were identified by their unique ultrastructural features including the presence of round synaptic vesicles and pale mitochondria (*Hammer et al., 2014*; *Bickford et al., 2010*). Retinal terminals clustering onto the same portion of a dendritic branch were classified as complex RG synapses if the membranes of terminals touched each other and were not isolated from each other by glial processes. On the other hand, a retinal terminal isolated from other retinal terminals was classified as a simple RG synapse. In each mouse, retinal terminals were identified regardless of their simple or complex designation and were then assigned to one of these two classes. The proportion of retinal terminals participating in each class was averaged from data sets obtained from the dLGN of three mice per genotype (2–3 data sets were obtained per mouse). T-test or ANOVA analysis were used to determine any significant difference of the mean values between groups.

The performance of a mouse in the training sessions was reported as the percentage of correct choices the mouse made out of 8 or 10 trials per day (e.g. day 1), and then an average of daily performances was calculated for each group of mice. The performance of a mouse in the test phase of both detection and discrimination tasks were reported as the percentage of correct choices the mouse made out of 10 trials per given task (e.g. for 10% contrast) and these values were used to calculate the mean for a group of mice (e.g. control group). T-test or ANOVA analysis were used to determine any significant difference of the mean values between groups.

## Acknowledgements

This work was supported in part by the National Institutes of Health – National Eye Institute grants EY021222 and EY024712 (MAF), an Independent Investigator grant from the Brain and Behavior Foundation (MAF), and a fellowship from the VTCRI Medical Research Scholars Program (AM). The authors are grateful to Drs. G Valdez, K Mukherjee and YA Pan for helpful comments and discussions. We also thank Drs. S Robel (VTCRI) and W Guido (U Louisville) for generously supplying *Aldh1l1-GFP* and C*rh-Cre* mice, respectively.

## Additional information

### Funding

| Funder | Grant reference number | Author |
| --- | --- | --- |
| National Eye Institute | EY021222 | Michael A Fox |
| Brain and Behavior Research Foundation | | Michael A Fox |
| National Eye Institute | EY024712 | Michael A Fox |
| Virginia Tech Carilion Research Institute | Medical Research Scholars Program | Aboozar Monavarfeshani |

The funders had no role in study design, data collection and interpretation, or the decision to submit the work for publication.

### Author contributions

Aboozar Monavarfeshani, Conceptualization, Formal analysis, Investigation, Methodology, Writing—original draft, Writing—review and editing; Gail Stanton, Jonathan Van Name, Kaiwen Su, William A Mills III, Kenya Swilling, Alicia Kerr, Jianmin Su, Formal analysis, Investigation; Natalie A Huebschman, Data curation, Formal analysis; Michael A Fox, Conceptualization, Resources, Formal analysis, Supervision, Funding acquisition, Investigation, Writing—original draft, Project administration, Writing—review and editing

Author ORCIDs

Aboozar Monavarfeshani http://orcid.org/0000-0001-8906-5115
Michael A Fox http://orcid.org/0000-0002-1649-7782

Ethics

Animal experimentation: All experiments were performed in compliance with National Institutes of Health (NIH) guidelines and protocols and were approved by the Institutional Animal Care and Use Committee (IACUC# 15-137VTCRI, 15-167VTCR and 15-174VTCRI) and Institutional Biosafety Committee (IBC# 15-038) at Virginia Tech.

Decision letter and Author response

Decision letter https://doi.org/10.7554/eLife.33498.019
Author response https://doi.org/10.7554/eLife.33498.020

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
