## [Decision Letter]

Thank you for submitting your manuscript "LRRTM1 underlies synaptic convergence in visual thalamus" to *eLife*. Your article has been reviewed by three peer reviewers, and the evaluation has been overseen by a Reviewing Editor, Jeremy Nathans, and a Senior Editor, Gary Westbrook. As you will see, all of the reviewers were impressed with the importance and novelty of your work.

I am including the three reviews at the end of this letter, as there are a variety of specific and useful suggestions in them. We appreciate that the reviewers' comments cover a range of suggestions for improving the manuscript. Please use your best judgment in deciding which of these can be accommodated in a reasonable period of time. In particular, the first suggestion from reviewer #1 ("It would be interesting to perform eye suturing experiments…") is an excellent idea but, is in our collective opinion beyond the scope of the present work. Most of the other reviewer comments represent relatively minor points that we believe can be accommodated.

Reviewer #1:

In this study, Monavarfeshani et al. have uncovered a molecular mechanism by which retinothalamic convergence occurs during development. It has recently been shown that a high degree of retinal convergence onto thalamic relay cells occurs in the dLGN of both rodents and higher mammals. Here, the authors describe a role for Leucine Rich Repeat Transmembrane Neuronal 1 (LRRTM1) expressed by dLGN relay cells in mediating this process. The expression is quite selective for dLGN relay cells, and this is a major strength of this study. The authors used multiple methods, including serial block-face scanning EM, to support their conclusions and also showed complex visual behavior deficits in lrrtm1 mutant mice. The authors demonstrate a role for lrrtm1 in retinothalamic convergence, thus providing an interesting addition to our understanding of the molecular underpinnings of this developmental phenomenon. Pending response to the issues raised below, this study appears suitable for publication in this journal.

1) What is the role of visual input on the development of retinothalamic convergence? It would be interesting to perform eye suturing experiments to determine if this phenomena still occurs at P14 even without visual input, strengthening the claim that lrrtm1 is required for convergence, especially since some clustering still occurs in lrrtm1 knockouts (Figure 5). This would not entail Serial Blockface EM, but one of the simpler approaches presented in this study and would really strengthen this present communication.

2) Do the authors have any later timepoints for the vGLUT2 staining and CTB experiments to demonstrate that the clustering is not simply delayed in the lrrtm1 mutants? (Related to Figure 5 and Figure 5—figure supplement 1)

3) At what timepoint are the brainbow AAVs being delivered for the tissue harvested at P8 and P10? I would expect that these need to be in utero injections, but it isn't clear in the Materials and methods section.

4) What developmental age are the retina ISH data in Figure 3?

5) Is there any explanation for the difference in GAD67 staining between Figure 4? Is the red arrow in Figure 4 pointing to a GAD67 positive cell? Please make this clear in the figure legend.

6) Can the authors provide quantification for the percent colocalization of lrrtm1 and nrn1 in relay cells (Figure 4)?

7) In Figure 6—figure supplement 1, it is not clear what conclusion one is supposed to draw about the connection between SBFSEM and CTB/vGLuT2 labeling. It is hard to make out a clear difference between controls and lrrtm1 mutants, as there is no quantification. Also, the word "Age" is written in the bottom left figure of Panel A, but it is not clear that this is supposed to appear there.

8) In Figure 5—figure supplement 1, what age is the retinal tissue that was used for staining?

9) Subsection “Identification of target-derived synaptic organizing molecules in dLGN”: molecules should be molecule.

10) Subsection “Impaired visual behaviors in mice lacking LRRTM1: performing task performance sounds awkward.

11) Subsection “Impaired visual behaviors in mice lacking LRRTM1”: behaviors task should be behavioral tasks.

12) Discussion section, clusters should be cluster.

13) Discussion section, cell should be cells.

14) Discussion section, ontogenetic should be optogenetic.

15) Subsection “LRRTM1 as a target-derived synaptic organizer in visual thalamus”: synapse should be synapses.

16) Subsection “Retinal convergence: artifact or by design?”: rely should be relay.

17) The first portion of the Discussion section is redundant with Introduction section of this paper - this can be edited extensively.

Reviewer #2:

In their submission entitled "LRRTM1 underlies synaptic convergence in visual thalamus", Monavarfeshani and colleagues first demonstrate that the formation of complex RG synapses in the dLGN begins between days P8 and P14 using a thorough battery of independent techniques. To investigate what signaling may underlie development of these synapses, they performed RNAseq experiments comparing the dLGN to the vLGN (where complex synapses do not form) over the course of development and identify lrrtm1 and nrn1 as differentially expressed candidate genes that have previously been implicated in excitatory synapse formation. Using multiple methods, they demonstrate that lrrtm1 mRNA and protein levels are increased at the appropriate developmental timepoints specifically in the dLGN and that this expression is specific to CRH positive relay neurons. To demonstrate a role of lrrtm1 in complex synapse formation, they use a knockout mouse to demonstrate that the formation of complex synapses is decreased when lrrtm1 is lost. Finally, to examine what visual functions may be mediated by complex synapses in the dLGN, visual function in lrrtm1 knockout mice is investigated using a forced swim task where mice discriminate between a visual stimulus of vertical bars and a gray screen, or horizontal bars. Knockout mice perform similarly to controls in regards to frequency and contrast detection thresholds when compared to a gray screen, but are significantly different from controls when tested against horizontal bars.

Overall this work is well done and is of appropriate interest to the community. Especially the use of multiple techniques to examine the formation and disruption of complex synapses in the dLGN. The recent understanding that there is significant retinal convergence in the LGN opens many questions as to the purpose of such circuit architecture. These experiments begin to elucidate the mechanisms of their formation in addition to what role they might play in visual processing.

I do believe that the behavioral aspects of this submission still require further clarification and experiments. To start, from my reading of the Materials and methods section, it is not very clear to me how the testing schedule in Figure 7 differ from Figure 7 and G. It appears as though mice in all experiments are trained using stimuli as in B and/or C for 8 days. Then in panels D and E they are tested on 1 of the 8 different conditions, once per day for 10 to 12 days. In panels F and G they are tested on the same stimulus 10-12 times for the indicated days. I think it would be best to further clarify the exact nature of these experiments since the effects on the "complex" visual task mediated by complex synapse loss is a crux of this papers findings. In addition, I am curious why the discrimination task (F and G) is tested on a divergent schedule from the detection task (D and E). This discrepancy makes it difficult to directly compare the two systems. While I appreciate the schedule in F and G to rule out known learning deficits in the lrrtm1 knockout, it would make a stronger case if the behaviors were tested similarly. Finally, the other difference between the detection and discrimination tasks is the availability of horizontal bars. It is plausible that the lrrtm1 knockout doesn't lead to an inability to discriminate, but instead effects detection of vertical bars differentially relative to horizontal bars. Thus, if experiments in panels 7D and E could be repeated putting horizontal bars against a gray background, or conversely testing panels F and G against a gray background, this would indicate a more clear role for discrimination itself as the trait lacking in lrrtm1 knockout mice. Conversely, if a difference is still seen moving panels F and G to a detection task, this might indicate the difference is due to the schedule of behavioral testing.

On the Cumulative Function graphs the y axis appears to pass beyond 100%. This makes it difficult to determine at what point on the x-axis the maximum has been reached. Perhaps rescale the y-axis appropriately or add a landmark on the graph to indicate 100%.

Would it be possible/helpful to expand the brainbow images into a Supplemental figure? It is quite hard to discern the discrete axons presented in some of the figures.

Viral titers used for the brainbow experiments should be indicated in the Materials and methods section.

Reviewer #3:

This is an interesting study that demonstrates a role for LRRTM1 in the maturation of retinal synapses in the dorsal lateral geniculate nucleus (dLGN). The interest arises from two sets of recent studies. One showed that retinal terminals differ in a target dependent manner, even when they likely arise from the same retinal cell types. Thus, LRRTM1 could be a factor that regulates branch-specific synaptogenesis. The second (reported in papers from Fox's group and two others) showed a greater degree of retinal convergence on dLGN neurons than had been supposed. LRRTM1 seems to selectively effect the complex synapses that mediate this convergence. Thus, although mechanistic analysis is lacking here, the result is quite important.

The results are for the most part fairly straightforward, and they are uniformly convincing and well documented. One confound is that the behavioral phenotype may not result exclusively from the LGN defect, but the authors have done all the controls they can, and are quite candid about the limitation. The only better approach would be a conditional mutant, but this would not straightforward because it would require an LGN-specific Cre line (the LRRTM seems to act postsynaptically) which may not be available.

The obviously valuable addition would be physiological tests of decreased convergence in the mutant. My own view is that there are few labs capable of this sort of recording and it is likely beyond the capabilities of the Fox lab, so there is no point in asking for it. I think the paper stands on its own, and likely others will take up the physiological challenge. I look forward to reading the other reviewers' opinion on this point.

Figure 1. Puncta size is nicely quantified from VGlut2 staining (panels C-H) but the authors also go on to document interareal differences using transgenic (I), EM (J) and viral (K-M) methods. It would be nice to see rudimentary quantification of at least one of these – perhaps Hb9-GFP would be most straightforward – to test whether the effects detected by the various methods are of similar magnitude.

Figure 1 and Figure 1—figure supplement 1 look remarkable similar. Please check to be sure there was not an inadvertent duplication.

Subsection “LRRTM1 is required for the development of complex RG synapses”. Having found that 10 and 63% of terminals are big in the two genotypes it is unnecessary to report that 90 and 37% are not big.

Subsection “LRRTM1 is required for the development of complex RG synapses”. Lrrtm1 loss includes both a decrease in the number of complex synapses as well as a decrease in the complexity of the remaining complex synapses. Is it possible to provide a single metric that includes both effects? (This percentage decrease could be included in the text; an additional figure is not necessary.)

Subsection “LRRTM1 is required for the development of complex RG synapses”. Although the focus is on synapse complexity, there is also an effect on synapse size (Figure 6). This is potentially important. Is there more to say about it? And shouldn't it be noted as a possible confound in the interpretation of the behavioral results?

Subsection “Impaired visual behaviors in mice lacking LRRTM1”. The description of the complex task in which the major effect was observed is somewhat cursory. More is needed. Do the panels in Figure 7 represent different sets of animals?

Discussion section. The first paragraph of the Discussion section on the recent discovery of convergence in LGN, is largely repetitive of the Introduction. One or the other could be reduced.

Figure 5—figure supplement 1 and Figure 7—figure supplement 1. The title claims more than is delivered in this figure.

---

## [Author Response]

Reviewer #1:1) What is the role of visual input on the development of retinothalamic convergence? It would be interesting to perform eye suturing experiments to determine if this phenomena still occurs at P14 even without visual input, strengthening the claim that lrrtm1 is required for convergence, especially since some clustering still occurs in lrrtm1 knockouts (Figure 5). This would not entail Serial Blockface EM, but one of the simpler approaches presented in this study and would really strengthen this present communication.

We agree that these are important questions, but also believe that eye-suturing experiments are beyond the scope of the current manuscript. However, we did test whether *lrrtm1* or *nrn1* mRNA expression was influenced by visual input. We measured mRNA levels in the dLGN of both light-deprived mice (using a late dark rearing model in which deprivation after P20 leads to a weakening of the strength of retinal inputs and an increase in the number of functional retinal inputs onto relay cells; Hooks and Chen, 2008) and transgenic mice which lack retinal projections to the brain (*math5*-/- mutants; Wang et al., 2001; Hammer et al., 2014). Our data (which we have added in Figure 3—figure supplement 1) show that the expression of *lrrtm1* and *nrn1* mRNAs are not significantly influenced by altering visual inputs to dLGN. These results may seem somewhat surprising given the established role of Nrn1 (a.k.a. CPG15) as an activity-regulated molecule in the developing brain. However, previous studies have shown that thalamic expression of *nrn1* is indeed insensitive to changes in neuronal activity, in contrast to levels in visual cortex (Corriveau et al., 1999).

2) Do the authors have any later timepoints for the vGLuT2 staining and CTB experiments to demonstrate that the clustering is not simply delayed in the lrrtm1 mutants? (Related to Figure 5 and Figure 5—figure supplement 1)

Yes, this is an important point. We have now added VGluT2 and CTB labeling of retinal terminal in adult control and mutant mice (60-85 days of age). Significant differences in the size of CTB- and VGluT2-labeled puncta exist in adult LRRTM1-deificient mice. These results have been added to Figure 5 and Figure 3—figure supplement 1. It is also important to point out that the SBFSEM analysis of mutant and control dLGN (in Figure 6) was performed at P42, an age in which retinogeniculate synapses appear adult-like anatomically and functionally.

3) At what timepoint are the brainbow AAVs being delivered for the tissue harvested at P8 and P10? I would expect that these need to be in utero injections, but it isn't clear in the Materials and methods section.

In order to label retinogeniculate terminals at P8, we performed AAV injections at P0. Surprisingly, the short period of infection (8 days) is sufficient in neonates for the expression and transport of fluorescent reporter proteins in RGCs. These details have been added to the Materials and methods section.

4) What developmental age are the retina ISH data in Figure 3?

These experiments were performed on P25 retina. This age has now been added to figure legend.

5) Is there any explanation for the difference in GAD67 staining between Figure 4? Is the red arrow in Figure 4 pointing to a GAD67 positive cell? Please make this clear in the figure legend.

This is a very astute observation. The difference is due to slight differences in the age of tissue imaged (P14 vs. P25). We have specified the difference in age in the image and have replaced the images with slightly better images. We have also added text in the figure legend that explains the red arrows indicate GAD67+ dLGN interneurons (and have added these arrows to both panels in Figure 4).

6) Can the authors provide quantification for the percent colocalization of lrrtm1 and nrn1 in relay cells (Figure 4)?

Quantifying the percent colocalization with all thalamic relay cells is quite difficult because tools to specifically label relay cells (and not other neurons) are not available. The *Crh-Cre* line labels only a subset of relay cells. For this reason, we quantified the total number of *syt1, gad1, lrrtm1*, and *nrn1* expressing cells in dLGN (by comparing to DAPI labeling). We then quantified the percent of *syt1*+, *gad1*+, or *Crh-Cre+* neurons that co-express either *lrrtm1* or *nrn1.* The important part is that 100% of the *Crh-Cre+* neurons express *lrrtm1* or *nrn1.* This data has been added to the Figure 3—figure supplement 1.

7) In Figure 6—figure supplement 1, it is not clear what conclusion one is supposed to draw about the connection between SBFSEM and CTB/vGLuT2 labeling. It is hard to make out a clear difference between controls and lrrtm1 mutants, as there is no quantification. Also, the word "Age" is written in the bottom left figure of Panel A, but it is not clear that this is supposed to appear there.

We have revised the figure legend to clarify this point. The point was to use a visual element to try to explain why VGluT2- and CTB-punta size decreases in mutants (by confocal analysis), but that at the EM level terminal size is not decreased but rather clusters are missing. This supplement was meant to help clarify this point, but if the reviewers prefer we can remove it.

The word “Age” has also been removed.

8) In Figure 5—figure supplement 1, what age is the retinal tissue that was used for staining?

This retinal tissue was from P12. This has been added to the figure legend.

9) Subsection “Identification of target-derived synaptic organizing molecules in dLGN”: molecules should be molecule.

Thank you. This has been fixed.

10) Subsection “Impaired visual behaviors in mice lacking LRRTM1”: performing task performance sounds awkward.

Thank you. This has been fixed.

11) Subsection “Impaired visual behaviors in mice lacking LRRTM1”: behaviors task should be behavioral tasks.

Thank you. This has been fixed.

12) Discussion section, clusters should be cluster.

Thank you. This has been fixed.

13) Discussion section, cell should be cells.

Thank you. This has been fixed.

14) Discussion section: ontogenetic should be optogenetic

Thank you. This has been fixed.

15) Subsection “LRRTM1 as a target-derived synaptic organizer in visual thalamus”: synapse should be synapses.

Thank you. This has been fixed.

16) Subsection “Retinal convergence: artifact or by design?”: rely should be relay.

Thank you. This has been fixed.

17) The first portion of the Discussion section is redundant with Introduction section of this paper-this can be edited extensively.This section has been revised extensively.Reviewer #2:I do believe that the behavioral aspects of this submission still require further clarification and experiments. To start, from my reading of the Materials and methods section, it is not very clear to me how the testing schedule in Figure 7 differ from Figure 7 and G. It appears as though mice in all experiments are trained using stimuli as in B and/or C for 8 days. Then in panels D and E they are tested on 1 of the 8 different conditions, once per day for 10 to 12 days. In panels F and G they are tested on the same stimulus 10-12 times for the indicated days. I think it would be best to further clarify the exact nature of these experiments since the effects on the "complex" visual task mediated by complex synapse loss is a crux of this papers findings.

Yes, this is correct. We have amended the Materials and methods section and Results section to better describe this approach and the results.

In addition, I am curious why the discrimination task (F and G) is tested on a divergent schedule from the detection task (D and E). This discrepancy makes it difficult to directly compare the two systems. While I appreciate the schedule in F and G to rule out known learning deficits in the lrrtm1 knockout, it would make a stronger case if the behaviors were tested similarly.

In fact, we started by using a similar schedule for the discrimination tasks but when we observed difficulty of performing these tasks (particularly in the mutants) we worried about whether learning deficits were confounding our results. We understand the confusion in using different paradigms. We have now included new data where we tested the discrimination task without also assessing task memory. We have added this data into Figure 7—figure supplement 1. We have added it to the supplement and not the main figure, because of the lower number of mutants and controls (n=6 per genotype) we had available at the time we received these reviews. While this analysis was performed on a limited number of animals, it did show a significant decrease in mutant performance in these tasks, supporting that data in Figure 7.

Finally, the other difference between the detection and discrimination tasks is the availability of horizontal bars. It is plausible that the lrrtm1 knockout doesn't lead to an inability to discriminate, but instead effects detection of vertical bars differentially relative to horizontal bars. Thus, if experiments in panels 7D and E could be repeated putting horizontal bars against a gray background, or conversely testing panels F and G against a gray background, this would indicate a more clear role for discrimination itself as the trait lacking in lrrtm1 knockout mice. Conversely, if a difference is still seen moving panels F and G to a detection task, this might indicate the difference is due to the schedule of behavioral testing.

We initially tried to address this concern (that the orientation of the gratings could impact performance) by showing that mutants, and controls for that matter, could learn to differentiate vertical bars from either gray screens or horizontal gratings with similar ability (see Figure 7). We now realize that what was missing from those analyses was the absence of vertical gratings, which could have biased our interpretation of those learning curves. To address this, we have now repeated those learning trials with a different cohort of mice (n=6 controls and 7 mutants), testing their ability to differentiate horizontal gratings (which were the positive cue) from a gray screen. Mutants and controls learned this task similar to how they learned to differentiate vertical gratings from gray screens or horizontal gratings. This new data is included in Figure 7—figure supplement 1 and is described in the text. Taken together, all of these results lead us to believe that LRRTM1 loss does not result in an altered ability to detect vertical versus horizontal bars.

The second part of this questions appears to suggest that we perform a contrast-sensitivity function for mutants and controls by challenging mice to a positive cue that changes both the spatial frequency and contrast of the gratings (while presenting a gray screen as a negative cue). We agree this is a good idea, but due to the number of animals required (especially to test both vertical and horizontal bars which would have to be done in 2 different cohorts of mice) and the time required, we believe these experiments are beyond the scope of the current manuscript.

On the Cumulative Function graphs the y axis appears to pass beyond 100%. This makes it difficult to determine at what point on the x-axis the maximum has been reached. Perhaps rescale the y-axis appropriately or add a landmark on the graph to indicate 100%.

These graphs have been rescaled as suggested.

Would it be possible/helpful to expand the brainbow images into a Supplemental figure? It is quite hard to discern the discrete axons presented in some of the figures.

We feel that these are critical for the manuscript and should remain in the main figures. However, we understand that some of the images are difficult to see the fine details so we have increased the size of the images in the revised version of the manuscript.

Viral titers used for the brainbow experiments should be indicated in the Materials and methods section.

These details have now been added to the methods of the revised manuscript.

Reviewer #3:Figure 1. Puncta size is nicely quantified from VGlut2 staining (panels C-H) but the authors also go on to document interareal differences using transgenic (I), EM (J) and viral (K-M) methods. It would be nice to see rudimentary quantification of at least one of these – perhaps Hb9-GFP would be most straightforward – to test whether the effects detected by the various methods are of similar magnitude.

We agree. We previously documented ultrastructural differences between retinal terminals in dLGN and vLGN using SBFSEM (Hammer et al., 2014). Here, we went back, as suggested, and analyzed terminal sizes in the dLGN and SC of HB9-GFP transgenic mice. A cumulative frequency distribution of GFP^+^ puncta size in dLGN and SC has now been added to Figure 1. Additionally, we added the average size of GFP^+^ puncta in the SC and dLGN of HB9-GFP mice to the revised text.

Figure 1 and Figure 1—figure supplement 1 look remarkable similar. Please check to be sure there was not an inadvertent duplication.

Thank you! You are absolutely correct and we have now fixed this mistake.

Subsection “LRRTM1 is required for the development of complex RG synapses”. Having found that 10 and 63% of terminals are big in the two genotypes it is unnecessary to report that 90 and 37% are not big.

We have revised this sentence to avoid redundancy.

Subsection “LRRTM1 is required for the development of complex RG synapses”. Lrrtm1 loss includes both a decrease in the number of complex synapses as well as a decrease in the complexity of the remaining complex synapses. Is it possible to provide a single metric that includes both effects? (This percentage decrease could be included in the text; an additional figure is not necessary.)

While we agree conceptually with this (to simplify the phenotypes) we have had trouble doing this without omitting important data. For example, one possibility is that we could report a single metric like we did for the brainbow analysis (Figure 6). However, this would not accurately demonstrate the decrease in complex RG synapse number in the mutants.

(Perhaps, it is important to note that for our AAV-brainbow analysis in mutants and controls did use only a single metric (by only looking at complex retinal terminals and classifying them into RG synapse with less or more than 4 terminals). The rationale for this was that we do not feel that it is scientifically valid to quantify a decrease in the number of complex RG synapses with a method (i.e. viral delivery of genes encoding fluorescent proteins) that does not label all retinal axons.)

Subsection “LRRTM1 is required for the development of complex RG synapses”. Although the focus is on synapse complexity, there is also an effect on synapse size (Figure 6). This is potentially important. Is there more to say about it? And shouldn't it be noted as a possible confound in the interpretation of the behavioral results?

We agree that this is a really remarkable phenotype and could be very important. We have now added additional data on this point, showing that active zones are increased in both simple and complex RG boutons in LRRTM1-deficient mutants. It is important to note, however, that when we normalize for terminal size the density of axon zones appears no different than in control boutons. All of this has been added to Figure 6. We have also added more details about the increase in bouton size in LRRTM1-deficient mutants in the discussion (and how this could confound interpretation of our results).

Subsection “Impaired visual behaviors in mice lacking LRRTM1”. The description of the complex task in which the major effect was observed is somewhat cursory. More is needed. Do the panels in Figure 7 represent different sets of animals?

We have added more detail to the methods and Results sections, as described above for questions from reviewer 2. Yes, different sets of animals we used for those experiments (Figure 7) and for the new data now added to Figure 6—figure supplement 1.

Discussion section. The first paragraph of the Discussion section on the recent discovery of convergence in LGN, is largely repetitive of the Introduction. One or the other could be reduced.

This has been revised.

Figure 5—figure supplement 1 and Figure 7—figure supplement 1. The title claims more than is delivered in this figure.

We agree and have not only changed the title but have split this figure into Figure 7—figure supplement 2 and Figure 2—figure supplement 1. This allowed us to provide additional, high magnification images of VGluT1^+^ and VGluT2^+^ nerve terminals in layer IV of visual cortex of mutants and controls.